# *Drosophila* TRPγ is required in neuroendocrine cells for post-ingestive food selection

**Subash Dhakal[1†], Qiuting Ren[2†], Jiangqu Liu[3†], Bradley Akitake[3], Izel Tekin[3], Craig Montell[3]\*, Youngseok Lee[1]\***

[1]Department of Bio and Fermentation Convergence Technology, Kookmin University, Seoul, Republic of Korea; [2]Department of Biological Chemistry, The Johns Hopkins University School of Medicine, Baltimore, United States; [3]Neuroscience Research Institute and Department of Molecular, Cellular and Developmental Biology, University of California, Santa Barbara, Santa Barbara, United States

**\*For correspondence:**
cmontell@ucsb.edu (CM);
ylee@kookmin.ac.kr (YL)

†These authors contributed equally to this work

**Competing interest:** The authors declare that no competing interests exist.

**Abstract** The mechanism through which the brain senses the metabolic state, enabling an animal to regulate food consumption, and discriminate between nutritional and non-nutritional foods is a fundamental question. Flies choose the sweeter non-nutritive sugar, L-glucose, over the nutritive D-glucose if they are not starved. However, under starvation conditions, they switch their preference to D-glucose, and this occurs independent of peripheral taste neurons. Here, we found that eliminating the TRPγ channel impairs the ability of starved flies to choose D-glucose. This food selection depends on *trpγ* expression in neurosecretory cells in the brain that express diuretic hormone 44 (DH44). Loss of *trpγ* increases feeding, alters the physiology of the crop, which is the fly stomach equivalent, and decreases intracellular sugars and glycogen levels. Moreover, survival of starved *trpγ* flies is reduced. Expression of *trpγ* in DH44 neurons reverses these deficits. These results highlight roles for TRPγ in coordinating feeding with the metabolic state through expression in DH44 neuroendocrine cells.

## Editor's evaluation

This manuscript reports the discovery of a role for the TRPγ channel in nutrient sensing behavior. The authors show that this gene functions in the Dh44+ cells to direct the animal to feed on nutritious sugar after fasting. Since the molecular mechanisms of nutrient sensing are still poorly defined, this manuscript presents a conceptual advance on this topic and also more broadly, on the role of TRP channels in organismal physiology.

## Introduction

In many animals, the decision about whether to feed on the most delectable food options versus the most nutritious options is influenced by the internal state. If they are sated, many animals ranging from flies to mammals select the most delicious foods, such as highly sweet forms of sugars, such as L-glucose, over the less sweet but nutritive D-glucose (*Burke et al., 2012*; *Burke and Waddell, 2011*; *de Araujo et al., 2008*; *Dus et al., 2015*; *Dus et al., 2011*; *Miyamoto et al., 2012*). This ability to detect the hedonistic quality of foods, including the sweetness of sugars, occurs through taste receptors in their peripheral taste organs. However, if an animal is starved, they alter their preference, and choose foods based on their caloric content.

The capacity to evaluate the nutrient quality of sugars can occur independent of any input from peripheral taste receptor cells. Mice missing the TRPM5 channel, which is critical for sensing sugars, retain the ability to sense caloric-rich sugars (*de Araujo et al., 2008*). Rather, these taste-blind animals detect nutritive sugar through neurons in the brain—the nucleus accumbens of the ventral striatum (*de Araujo et al., 2008*).

The fruit fly, *Drosophila melanogaster*, is also endowed with neurons in the brain that enable starved animals to use a post-ingestive mechanism to choose nutrient-rich sugars over their non-metabolizable, but sweeter enantiomers. In support of this finding, starved flies undergo a change in preference for D-glucose over L-glucose, and will do so in the absence of sugar receptors expressed in peripheral gustatory receptor neurons. Their ability to switch their preference to nutritive food depends on a small cluster of six neurons in the brain that express diuretic hormone 44 (DH44)—a neuropeptide related to the mammalian corticotropin-releasing hormone. These neuroendocrine cells located in a region of the brain called the pars intercerebralis (PI) are activated by nutritive sugars (*Dus et al., 2015*). DH44 neurons extend axons from the PI to the gut (*Dus et al., 2015*). The fly gut includes the crop, which is the equivalent of the mammalian stomach, and the intestines, which control feeding intake, food absorption, and defecation.

To characterize the molecular mechanism underlying taste-independent sensation of the caloric content of sugar, we tested potential contributions of transient receptor potential (TRP) channels. These channels are expressed in many peripheral sensory neurons, and function in phototransduction, chemosensation, mechanosensation, and temperature sensation. In addition, many TRP channels are expressed in the brain (*Cornillot et al., 2019*; *Venkatachalam and Montell, 2007*). Here, we report that *trpγ* is expressed in DH44 neuroendocrine cells in the PI. Mutations that eliminate the TRPγ cation channel disrupt the ability of food-deprived flies from shifting their preference from the highly palatable L-glucose to the nutritive D-glucose. In the absence of TRPγ, we observed a host of related metabolic defects, including a reduction in intracellular sugar levels, and diminished glycogen stores under starvation conditions. These deficits were due to loss of *trpγ* expression in DH44 neuroendocrine cells. Our findings highlight a key role for TRPγ in taste-independent food selection, and the metabolic response to nutritive sugars under starvation conditions.

## Results

### TRPγ is required for choosing nutrient-rich glucose in starved flies

Sated flies prefer the sweeter but non-nutritive L-glucose over the nutritive sugar D-glucose (*Dus et al., 2011*; *Miyamoto et al., 2012*; *Stafford et al., 2012*). However, starved flies switch their preference to D-glucose (*Dus et al., 2011*; *Miyamoto et al., 2012*; *Stafford et al., 2012*). The preference in sated flies depends on the taste sensors in the fly tongue, the labellum, while a taste-independent system in the brain functions in the selection of D-glucose in starved flies (*Dus et al., 2015*; *Dus et al., 2011*; *Miyamoto et al., 2012*; *Stafford et al., 2012*).

In addition to expression in peripheral sensory neurons, many TRP channels in mammals are expressed in internal tissues that function in sensing the internal metabolic state (*Dhakal and Lee, 2019*). Therefore, to address whether a *Drosophila* TRP channel contributes to the post-ingestive pathway used by starved flies to choose D-glucose, over L-glucose, we conducted two-way choice assays. We selected 200 mM L-glucose and 50 mM D-glucose for these assays since control flies (*w*[1118]) that were restricted from eating for a relatively brief period (5 hr) selected L-glucose (*Figure 1A*). These flies are referred to as 'sated', but were prevented from feeding for several hours to motivate feeding. However, if the flies were starved for a prolonged period (18 hr), they selected the metabolizable 50 mM D-glucose over non-metabolizable 200 mM L-glucose (*Figure 1B*; *Dus et al., 2015*; *Dus et al., 2011*; *Miyamoto et al., 2012*; *Stafford et al., 2012*). The change in food preference between 5 and 18 hr has been documented previously and is consistent with the observation that sugar levels in the hemolymph remain high after 5 hr of starvation, but drop significantly after 15 hr or more of starvation (*Dus et al., 2013*). We refer to flies that have been food restricted for 18 hr as 'starved'.

*Drosophila* encodes 13 members of the TRP superfamily of cation channels. Flies with mutations in any of 11 genes encoding TRP channels are homozygous viable and healthy, while mutations in *trpM* or *nompC* either cause pupal lethality or minimal survival as adults. We tested all 11 of the homozygous viable mutations to determine if any disrupted the switch in preference to D-glucose

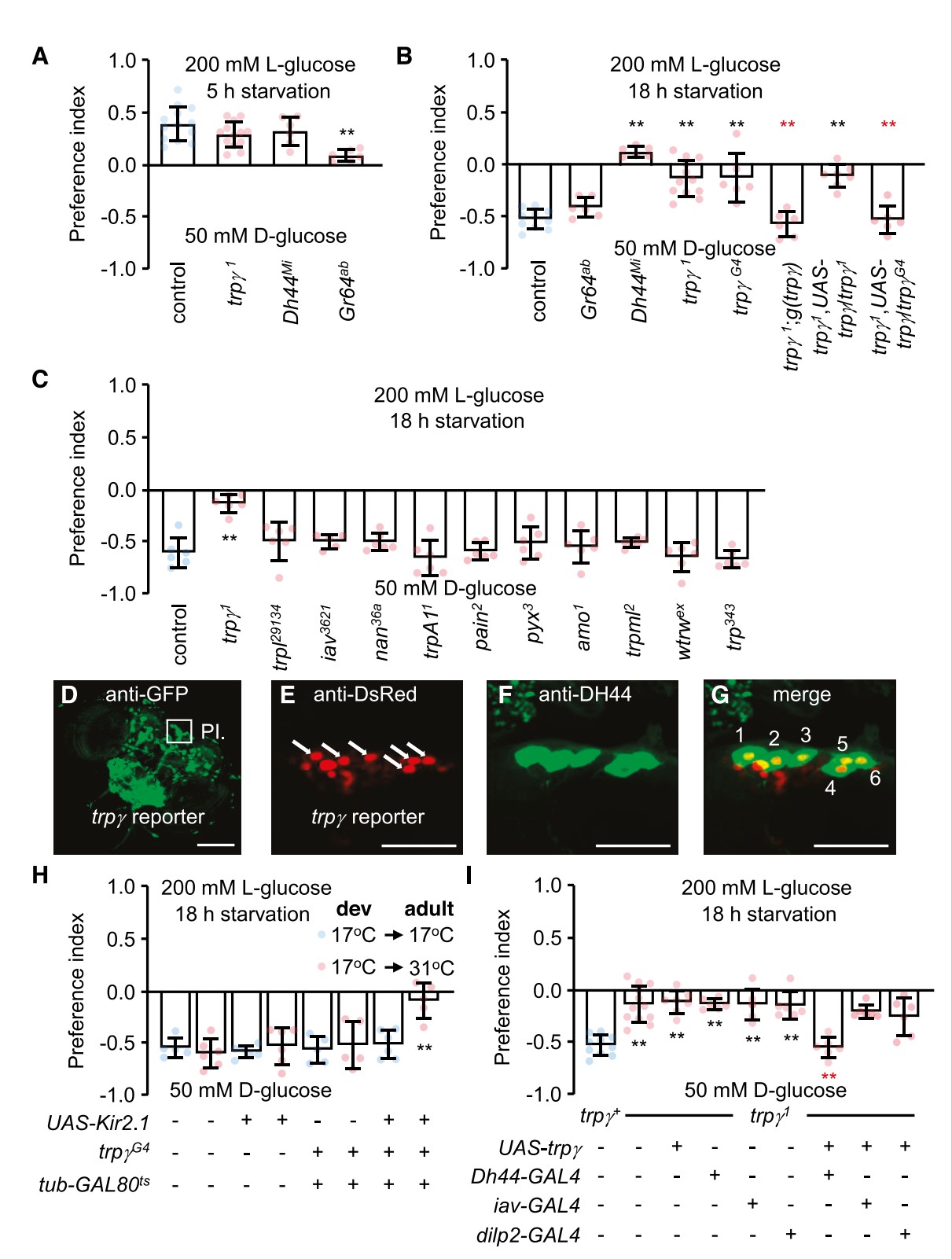

**Figure 1.** Characterization of TRPγ as a post-ingestion sensor using two-way feeding assays. (**A**) Binary food choice assay showing preferences of control (*w1118*) and indicated flies (both males and females) to nutritive sugar (50 mM D-glucose) versus non-nutritive, but sweeter sugar (200 mM L-glucose) after 5 hr starvation. n = 6–13. (**B**) Binary food choice assay with the same food in A with the indicated flies after 18 hr starvation. All data points include a mixture of males and females. n = 6–11. (**C**) Screening *trp* mutants after 18 hr starvation using binary food choice assays (50 mM D-glucose versus

*Figure 1 continued on next page*

*Figure 1 continued*

200 mM L-glucose). All data points include a mixture of males and females. n = 6. (**D**) Expression of a *trpγ-GAL4* (*trpγ^G4^/+*) reporter in brains. Whole brains from a *trpγ^G4^/+ > UAS-*mCD8::GFP fly were stained with anti-GFP. The pars intercerebralis (PI) is indicated by the square. Scale bar represents 25 μm. (**E–G**) Co-staining of a brain from a *trpγ^G4^/+;UAS-nls::tdtomato/+* fly with (**E**) anti-DsRed (*trpγ* reporter) and (**F**) anti-diuretic hormone 44 (DH44). (**G**) Merged image of (**E**) and (**F**). Numbers (1–6) indicate cells stained with anti-DsRed and anti-DH44. Scale bars in E–G indicate 25 μm. (**H**) Binary food choice assay with the indicated flies starved for 18 hr. 17°C and 31°C are the permissive and non-permissive temperature for the temperature-sensitive GAL4 repressor (GAL80^ts^), respectively. The flies were either *w^1118^* or *w^1118^* with the indicated transgenes. All were cultured during development (dev) at 17°C until eclosion, and then maintained at 17°C during adulthood (adult) or switched to 31°C. All data points include a mixture of males and females. n = 5–6. (**I**) Binary food choice assays to test for rescue of the defect in selecting D-glucose over L-glucose in *trpγ^1^* flies (males and females) that were starved for 18 hr. n = 6–12. Means ± SEMs. The black and red asterisks indicate significant differences from the controls and mutants, respectively (**p < 0.01) using ANOVA with Scheffe's analysis as a post hoc test or unpaired Student's t-tests (**H**). The red asterisks indicate significant levels of rescue.

The online version of this article includes the following figure supplement(s) for figure 1:

**Figure supplement 1.** Tip recordings, and behavioral assays indicating that *trpγ* functions in diuretic hormone 44 (*Dh44*) neurons, and not in peripheral neurons.

after prolonged starvation. As previously shown (*Dus et al., 2015*), this taste-independent pathway depends on DH44 neurons in the brain that express the neuropeptide DH44, but not on peripheral sugar receptors such as GR64a (*Figure 1B*). Ten of the 11 mutants tested displayed similar preferences for D-glucose as control flies (*Figure 1C*). In contrast, mutation of *trpγ* (*trpγ^1^*) virtually eliminated the bias for D-glucose in starved flies (*Figure 1C*). However, the *trpγ^1^* flies preferred L-glucose if they were sated (*Figure 1A*). Consistent with previous reports, the preference for L-glucose in sated flies is not dependent on DH44, but is reduced upon mutation of sugar receptors, which are expressed in gustatory receptor neurons in the periphery (*Figure 1A*).

The impairment in selecting D-glucose by the starved *trpγ^1^* flies was due to mutation of *trpγ* rather than a background mutation since a second *trpγ* allele (*trpγ^G4^*) exhibited the same phenotype (*Figure 1B*). Moreover, we rescued the deficit in choosing D-glucose with a genomic *trpγ^+^* transgene, g(*trpγ*), and with a *UAS-trpγ* cDNA expressed under control of the *GAL4* knocked into the *trpγ* locus (*trpγ^G4^* flies; + ). The *trpγ* mutant phenotype did not appear to be due to a deficit in the peripheral gustatory receptor neurons that are associated with sensilla decorating the fly tongue—the labellum. Using tip recordings on sugar-activated L6 sensilla, we found that the *trpγ^1^* mutants exhibited similar frequencies of D-glucose- and L-glucose-induced action potentials as control flies (*Figure 1—figure supplement 1A, B*). In addition, we performed proboscis extension response (PER) assays by applying D-glucose or L-glucose to the labellum. Both sated and starved *trpγ^1^* mutants displayed the same attraction to these sugars as control flies (*Figure 1—figure supplement 1C, D*). The time to full extension of the proboscis was also similar between the *trpγ^1^* mutant and the control, and we did not detect any obvious motor deficit in proboscis extension by the *trpγ^1^* mutants (*Figure 1—figure supplement 1E*; *Video 1* and *Video 2*).

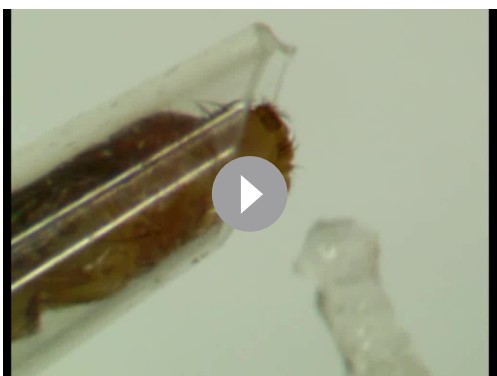

**Video 1.** Control fly displaying a proboscis extension response (PER) to 200 mM L-glucose after 18 hr starvation.

https://elifesciences.org/articles/56726/figures#video1

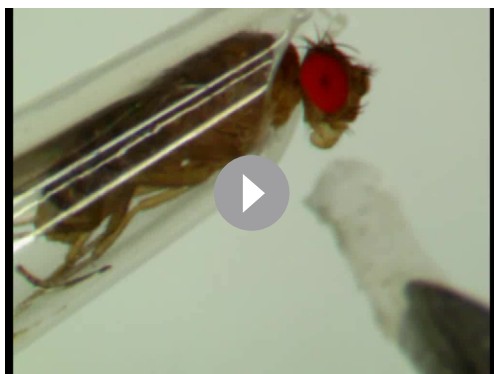

**Video 2.** *trpγ^1^* mutant fly displaying a proboscis extension response (PER) to 200 mM L-glucose after 18 hr starvation.

https://elifesciences.org/articles/56726/figures#video2

## TRPγ is required in DH44 neuroendocrine cells in the pars intercerebralis

To provide insight into the cellular basis for the taste-independent requirement for TRPγ for selecting nutrient-rich D-glucose under starvation conditions, we examined the *trpγ* expression pattern. Key candidate neurons for expressing *trpγ* are six cells in the pars intercerebralis (PI), which are located in dorsal medial area of the brain and comprise a set of taste-independent neurons that are important for enabling starved flies to switch their preference to D-glucose (*Dus et al., 2015*). These neurons express the neuropeptide, DH44, which is related to the mammalian corticotropin-releasing hormone, and are activated by nutritive sugars (*Dus et al., 2015*).

To explore whether *trpγ* is expressed in DH44 neuroendocrine cells in the brain, we took advantage of the *GAL4* reporter (*trpγ^G4^/+*), which we knocked into the *trpγ* locus (*Akitake et al., 2015*). The *trpγ* reporter was expressed in a large collection of neurons in the brain, including the PI (*Figure 1D*). Moreover, all six of the neurons that most strongly expressed the *trpγ* reporter were stained with anti-DH44 (*Figure 1E–G*). Since there are only six DH44-positive neurons in the PI (*Dus et al., 2015*), our finding indicates that all DH44 neurons in the PI express *trpγ*.

DH44-positive neurons are required in starved flies for selecting nutritive D-glucose over non-nutritive L-glucose (*Dus et al., 2015*). As previously shown (*Dus et al., 2015*), inactivating DH44 neurons by expressing a genetically encoded inwardly rectifying K$^+$ channel (*UAS-Kir2.1*) (*Nitabach et al., 2002*) under control of the brain-specific *Dh44-GAL4* (*Dus et al., 2015*) impairs the preference for D-glucose (*Figure 1—figure supplement 1F*). Some neurons in the PI express dILP2; however, inactivation of dILP2 neurons did not have any effect (*Figure 1—figure supplement 1F*). Inactivation of all *trpγ* neurons with *UAS-Kir2.1* using the *trpγ* reporter (*trpγ^G4^/+*) caused lethality during the pupal stage, presumably due to the broad expression in the brain. Therefore, to prevent *UAS-Kir2.1* expression during development in *trpγ^G4^/+* flies, we used a transgene that encodes a temperature-sensitive GAL4 repressor (GAL80$^{ts}$) that is expressed widely under control of the tubulin promoter (*McGuire et al., 2004*). GAL80$^{ts}$ is active at 17°C and suppresses GAL4 transcriptional activity. At 31°C GAL80$^{ts}$ is inactive, and therefore ineffective at suppressing GAL4. We found that when we maintained *trpγ^G4^/UAS-Kir2.1,tub-GAL80^{ts}* flies at 17°C, which prevented Kir2.1 expression, the starved animals showed a normal preference for the nutritious D-glucose (*Figure 1H*). However, when we reared the flies at 17°C until they were 3-day-old adults, and then switched them to 31°C for 18 hr prior to performing the binary food choice assays, the starved animals showed the same deficit in choosing the D-glucose over L-glucose as *Dh44-GAL4 > UAS-Kir2.1* flies (*Figure 1H*, *Figure 1—figure supplement 1F*).

To test if *trpγ* expression in DH44 cells is sufficient to restore normal nutrient selection to hungry *trpγ* mutant flies, we expressed the wild-type *trpγ* transgene (*UAS-trpγ*) under control of the *Dh44-GAL4* in the *trpγ¹* mutant background (*Figure 1I*). In addition, since *trpγ* is required in chordotonal neurons for coordination (*Akitake et al., 2015*), we also tested for rescue using the *iav-GAL4*, which directs expression in chordotonal neurons. We found that the preference for D-glucose was fully recovered in *trpγ* mutant flies by expression of *trpγ* in DH44 neuroendocrine cells, but not in chordotonal neurons (*Figure 1I*). To test further the requirement for *trpγ* in DH44 cells, we performed RNAi-mediated knockdown. We found that knockdown of *trpγ* in DH44 cells significantly impaired D-glucose selection, while flies harboring just the *Dh44-GAL4* or the *UAS-trpγ^RNAi^* transgenes displayed normal preference (*Figure 1—figure supplement 1G*).

The rescue results and the ability to phenocopy the *trpγ* deficit by RNAi using the *Dh44-GAL4* support our conclusion that *trpγ* is required and sufficient in DH44 neurons for proper selection of nutritive D-glucose in hungry flies. Additional support for this conclusion is our finding that the *trpγ* reporter is expressed in DH44 neurons. The validity of *trpγ* expression in DH44 neurons is also supported by the design of the *GAL4* driver, which is inserted precisely at the site of the initiation codon (*Akitake et al., 2015*). However, we do not have independent verification that the reporter fully recapitulates the native expression pattern of *trpγ*.

Due to the evidence that *trpγ* functions in DH44 neuroendocrine cells, we examined the effects of loss of *trpγ* on D-glucose-induced Ca$^{2+}$ dynamics in DH44 neurons in ex vivo brains from starved flies. To conduct this analysis, we expressed *UAS-GCaMP6s* under control of the *Dh44-GAL4* in control and *trpγ¹* animals. As previously shown (*Dus et al., 2015*), D-glucose, but not L-glucose, induces a Ca$^{2+}$ rise in control flies (*Figure 2A–C*). The DH44 neurons from *trpγ¹/+* heterozygous flies showed a

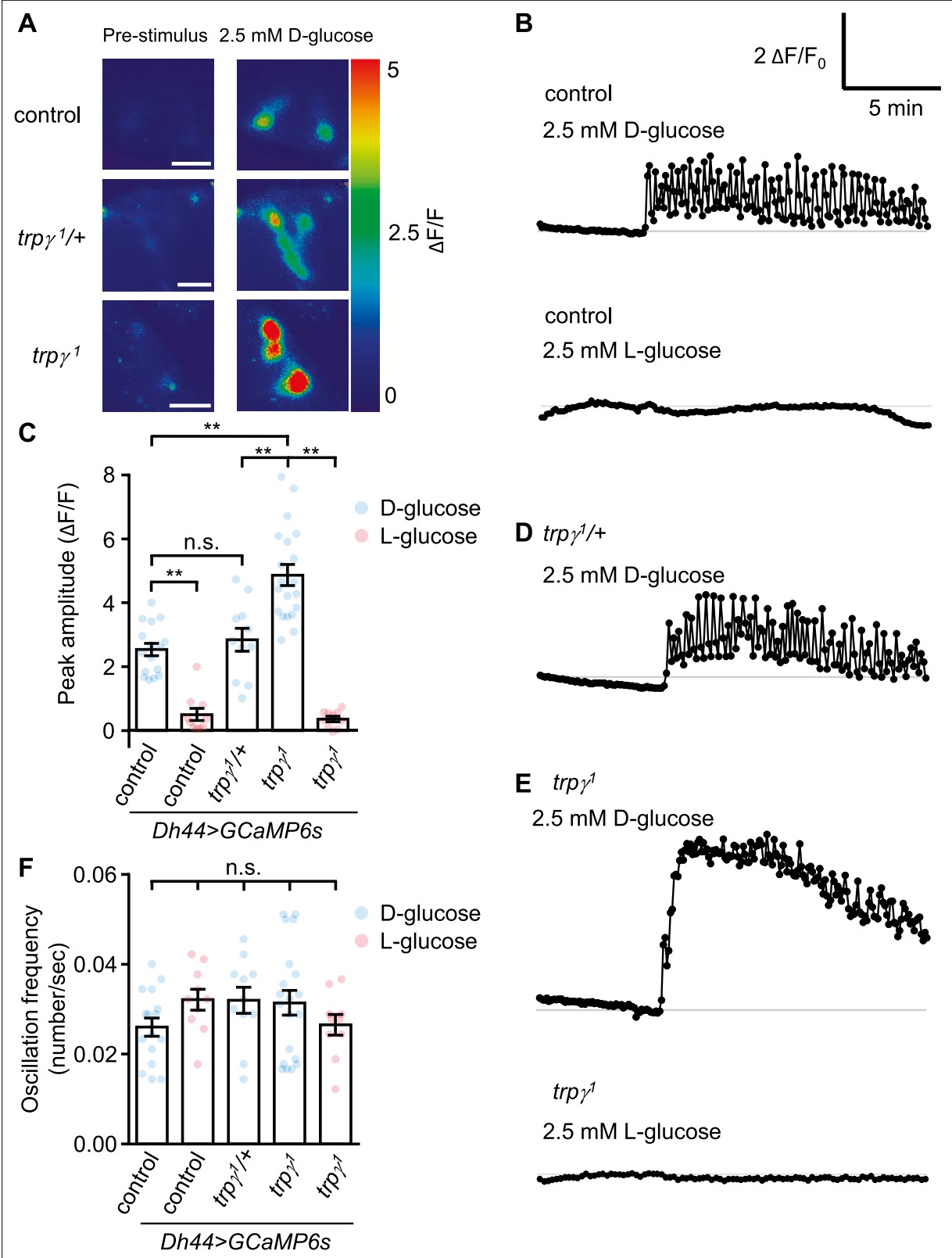

**Figure 2.** GCaMP responses of diuretic hormone 44 (*Dh44*) neurons to D-glucose or L-glucose. Changes in GCaMP6s fluorescence in *Dh44* neurons exposed to D-glucose or L-glucose. Brains were isolated from control, *trpγ¹/+*, and *trpγ¹* male flies carrying the *Dh44-GAL4* and *UAS-GCaMP6s*, and exposed to 2.5 mM D-glucose or L-glucose. (**A**) Images showing changes in GCaMP6s fluorescence. The left images show the pre-stimulation Ca²⁺ signals and the right images show the responses to 2.5 mM D-glucose. Scale bar, 20 µm. (**B**) Representative traces of *Dh44* neuronal responses to D-

*Figure 2 continued on next page*

*Figure 2 continued*

glucose or L-glucose from control brains. (**C**) Quantification of GCaMP6s responses to 2.5 mM D-glucose and 2.5 mM L-glucose. The peak amplitude ($\Delta$F/F) was calculated by subtracting the pre-stimulation baseline from the sugar-evoked peak value. The pre-stimulation baselines were calculated as the average of 10 frames before adding the sugar. n = 10–20 cells derived from 3–4 flies per genotype. (**D**) Representative trace of GCaMP6s response to D-glucose in *trpγ¹/+*. (**E**) Representative traces of GCaMP6s responses to D-glucose and L-glucose in *trpγ¹*. (**F**) Quantification of the oscillation frequencies in GCaMP6s responses to D-glucose and L-glucose. n = 10–20 cells. Means ± SEMs. The asterisks indicate significant differences (\*\*p < 0.01) using the Mann-Whitney test; n.s., not significant.

The online version of this article includes the following figure supplement(s) for figure 2:

**Figure supplement 1.** Quantification of GCaMP6s responses of diuretic hormone 44 (DH44) neurons to stimulation with 20 mM D-glucose.

similar $Ca^{2+}$ rise and oscillation frequency as the controls (*Figure 2A–D and F*). Surprisingly, we found that the peak $Ca^{2+}$ responses were higher in *trpγ¹* DH44 cells when we stimulated with 2.5 mM D-glucose (*Figure 2A, C and E*) and 20 mM D-glucose (*Figure 2—figure supplement 1*). However, the oscillation frequencies were similar to control flies (*Figure 2B, C, E and F*), and there was not a large difference in the rate of decline from the peak fluorescence (*Figure 2B and E*; time for 50% decline, $t_{50}$=10.1 and 8.2 min for the control and *trpγ¹*, respectively).

Monitoring changes in $Ca^{2+}$ signals is a very useful but imperfect proxy for neuronal activation since it is possible to observe a rise in $Ca^{2+}$ that is not associated with an increase in neuronal activity. Therefore, we used a genetically encoded voltage indicator, ASAP2s (*Chamberland et al., 2017*), in conjunction with fast two-photon imaging to record dynamic, rapid changes in voltage. To conduct these experiments, we expressed *UAS-ASAP2s* under the control of *Dh44-GAL4* in both control and *trpγ¹*. We found that when we bathed the brains in a glucose-free buffer, the basal voltage signals were significantly higher in DH44 neuroendocrine cells from the *trpγ¹* mutant than from control cells (*Figure 3A*). Furthermore, when we switched to a $Ca^{2+}$-free buffer, the control DH44 cells displayed elevated ASAP2s signals similar to the levels produced in *trpγ¹* in a $Ca^{2+}$-containing buffer (*Figure 3A*). These data indicate that $Ca^{2+}$ influx is essential to maintain an optimal basal level of neuronal activity.

Next, we measured the voltage responses by DH44 neurons in response to glucose. Stimulation of control cells with 20 mM D-glucose increased the peak amplitude 2.3-fold relative to the buffer only (*Figure 3B and D*). However, *trpγ¹* cells exposed to D-glucose, displayed an increase in the peak ASAP2s amplitude that was significantly larger than the control (*Figure 3B, D and F*). In contrast, L-glucose had no effect on the ASAP2s signal in DH44 neurons from either the control or *trpγ¹* (*Figure 3B*). Furthermore, the peak amplitude in the *trpγ¹* was similar to the responses of the control and *trpγ¹* under $Ca^{2+}$-free conditions (*Figure 3B and G*). However, the response latencies exhibited by the control and *trpγ¹* cells exposed to a $Ca^{2+}$-containing buffer or $Ca^{2+}$-free buffer were not significantly different (*Figure 3C*). Altogether, these data indicate that $Ca^{2+}$ influx through the TRPγ channel serves to attenuate rather than increase basal and glucose-stimulated activity of DH44 neurons.

The higher voltage responses exhibited by the *trpγ¹* DH44 neurons might impair nutrient sensing behavior by interfering with release of the DH44 neuropeptide. To address this hypothesis, we starved the flies for 18 hr, dissected their brains, incubated them with D-glucose, L-glucose, or D-fructose, and stained the cells with anti-DH44. Consistent with previous findings (*Dus et al., 2015*), we found that the DH44 cells from control flies exhibited lower anti-DH44 signals when stimulated with nutritious sugars (D-glucose and D-fructose), but not when bathed with the non-nutritious sugar (L-glucose) (*Figure 3—figure supplement 1A-C*). The diminished anti-DH44 staining was more pronounced when stimulated with 2.5 and 20 mM D-glucose than with 50 mM D-glucose. Of significance, we found that the anti-DH44 signals were lower in *trpγ¹* than in control DH44 neurons, regardless of whether or not the brain was activated by nutritious or non-nutritious sugar. The lower levels even in non-stimulated cells may reflect the higher basal activity of DH44 cells in the *trpγ¹* mutant, resulting in greater DH44 release even in the absence of stimulation.

## TRPγ is essential for regulating tissue sugar levels

Circulating sugar levels in the hemolymph consist mainly of trehalose and glucose, and decline during periods of starvation (*Dus et al., 2013*). The sugar concentration in the hemolymph of either fed or starved *trpγ¹* and *Dh44^Mi* mutant flies were indistinguishable from control animals (*Figure 4A*), indicating that higher circulating sugar levels do not provide an explanation for the impaired preference for nutritive D-glucose in the starved *trpγ¹* and *Dh44^Mi* flies. However, the cellular sugar levels in *trpγ¹*

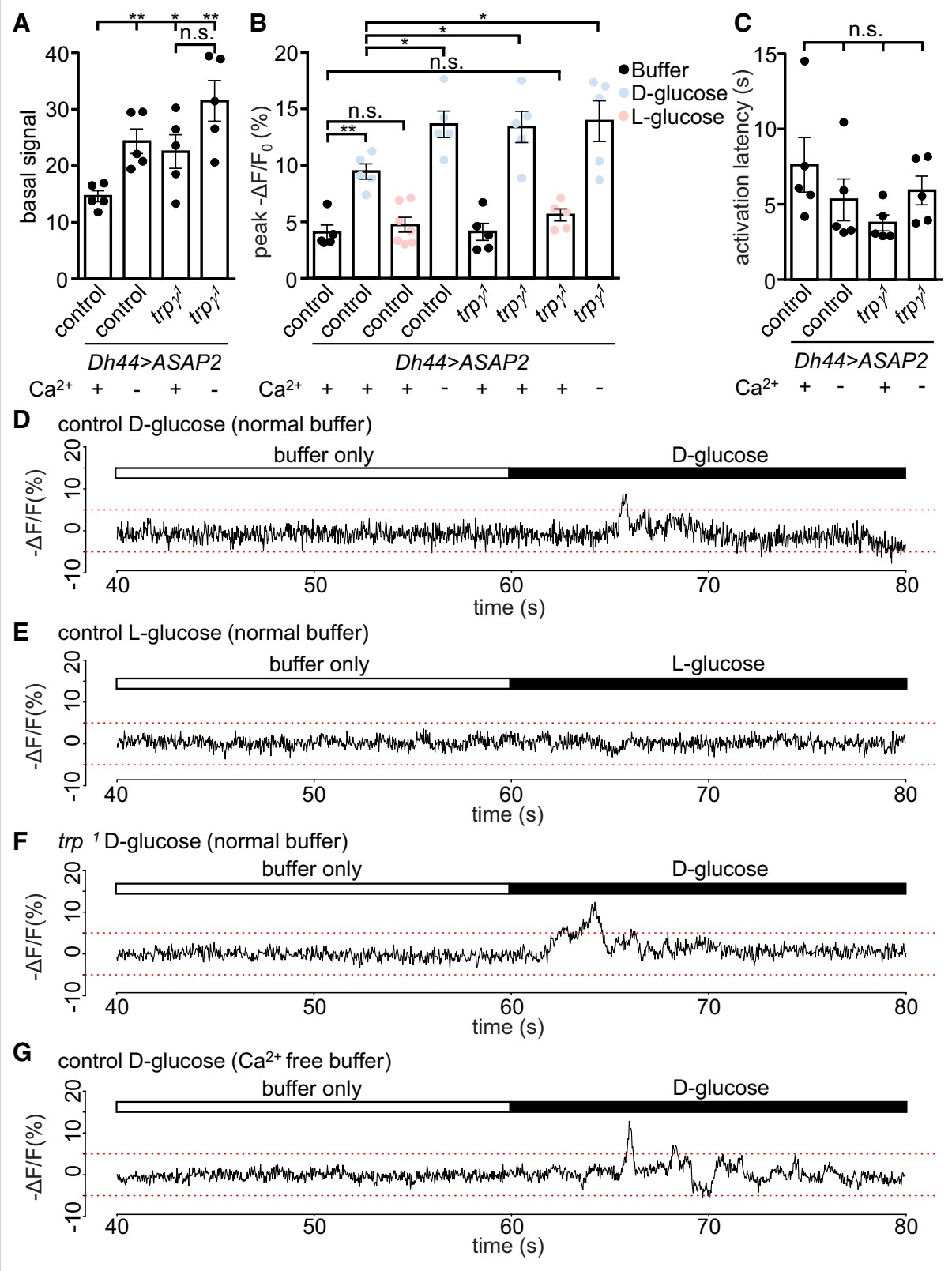

**Figure 3.** Voltage imaging of diuretic hormone 44 (DH44) neurons with ASAP2s. The *Dh44-Gal4* and *UAS-ASAP2s* transgenes were introduced into the control and *trpγ* [1] backgrounds. The DH44 neurons were activated by D-glucose or L-glucose in ex vivo brain preparations. (A–C) Voltage responses from brain preparations treated with normal (Ca$^{2+}$-containing) buffer alone, normal buffer containing either 20 mM D-glucose or L-glucose, or Ca$^{2+}$-free buffer alone. (A) Quantification of peak basal ASAP2s signals in normal buffer (+) or Ca$^{2+}$-free buffer (-) in control or *trpγ* [1] DH44 neurons. The peak basal

*Figure 3 continued*

signals were calculated as the maximum value of brightness for 20 s (arbitrary values obtained using Bruker Prairie View 5.5), prior to the application of the stimuli. n = 5 brains. One neuron was analyzed per brain. The asterisks indicate significant differences from the control in $Ca^{2+}$-containing buffer. (B) Quantification of peak voltage amplitude of DH44 neurons in response to D-glucose and L-glucose. The presence (+) or absence (-) of $Ca^{2+}$ in the buffer is indicated. The peak amplitudes (-$\Delta F/F$) were calculated by subtracting the pre-stimulation baseline from the stimulus-evoked peak value. The pre-stimulation baselines were calculated as the average of 100 data points (corrected for photobleaching) before stimulation. n = 5–7 brains. One neuron was analyzed per brain. (C) Quantification of the activation latencies of DH44 neurons in normal buffer (+) or $Ca^{2+}$-free buffer (-). The activation latencies were calculated as the time (s) before the first activation peak occurred following application of the stimulus. n = 5 brains. One neuron was analyzed per brain. The analyses of statistical significance were based on comparisons with the control in $Ca^{2+}$-containing buffer. (D–G) Representative traces of results quantified in B. (D) Response of DH44 neurons from control brains to 20 mM D-glucose. (E) Response of DH44 neurons from control brains to 20 mM L-glucose. (F) Response of DH44 neurons from *trpγ¹* brains to 20 mM D-glucose. (G) Response of DH44 neurons from control brains to 20 mM D-glucose in a $Ca^{2+}$-free buffer. Means ± SEMs. The asterisks indicate significant differences using single-factor ANOVA with Scheffe's analyses between each test condition and the control. *p < 0.05, **p < 0.01, n.s., not significant.

The online version of this article includes the following figure supplement(s) for figure 3:

**Figure supplement 1.** Quantification of intracellular anti-diuretic hormone 44 (DH44) signals in DH44 neurons after stimulation with different concentrations of D-glucose, L-glucose, and D-fructose.

or *trpγ^G4* flies were significantly reduced under both fed and starved conditions (*Figure 4B*), and this phenotype was similar in the *Dh44* mutants (*Figure 4B*). We rescued these impairments in the *trpγ¹* mutant with the wild-type *trpγ* genomic transgene (*UAS-trpγ*) expressed under control of the *trpγ-GAL4* (*trpγ^G4*), and the *Dh44-GAL4*, but not with the *dilp2-GAL4* (*Figure 4B*). Furthermore, knockdown of *trpγ* in DH44 cells significantly reduced cellular sugar levels under fed and starved conditions, while flies harboring just the *Dh44-GAL4* or the *UAS-trpγ^RNAi* transgenes displayed normal sugar levels (*Figure 4—figure supplement 1A*).

Our findings suggest that the key deficit in *trpγ* mutants is glucose uptake. However, glucose uptake has not been assayed in flies, and we were also unable to develop such measurements. Consistent with reduced cellular sugar levels, the glycogen stores were also diminished in *trpγ* and *Dh44* mutant flies (*Figure 4C*). We rescued the reduction in glycogen levels in *trpγ* mutants by expressing *UAS-trpγ* in DH44 neurons, but not *dilp2* neurons (*Figure 4C*). In addition, RNAi-mediated knockdown of *trpγ* in DH44 cells significantly decreased glycogen stores under fed and starved conditions (*Figure 4—figure supplement 1B*).

The findings that *trpγ* is required under starvation conditions for proper metabolism raises the possibility that the mutant flies might have decreased survival when they are restricted from feeding. If we starve control flies, 50% die ($LT_{50}$) after 46.0 ± 2.1 hr (*Figure 4—figure supplement 1C*). However, the *trpγ* mutants are not able to survive as long in the absence of food (*Figure 4—figure supplement 1C*; $LT_{50}$: *trpγ¹*, $LT_{50}$ = 34.0 ± 2.9 hr, and *trpγ^G4*, $LT_{50}$ = 30.0 ± 3.2 hr). We rescued this phenotype using the *Dh44-GAL4* to drive expression of *UAS-trpγ* (*Figure 4—figure supplement 1C*; $LT_{50}$ = 49.5 ± 3.2).

## TRPγ is necessary for limiting consumption

DH44 neurons extend processes from the PI that label the gut and crop (*Dus et al., 2015*). Similarly, we found that the *trpγ* reporter stained neuronal varicosities in the epithelium of the crop and intestine (*Figure 5A and B*). Due to co-expression of *trpγ* with DH44 in the PI, these neuronal processes are likely to also extend from the PI. The crop serves to receive and store liquid food, before it is passed to the ventriculus for digestion. Expression in the crop suggests that *trpγ* might contribute to feeding since the state of satiation is monitored in part by receptors, which sense extension of these internal organs.

To test whether mutation of *trpγ* causes a feeding impairment, we transferred flies from regular food to vials containing 5% sucrose spiked with blue food dye, and determined the amount ingested over time. The food consumed by control flies peaks after 15 min (*Figure 5C*). We found that the levels of consumed food were elevated in *trpγ¹* relative to control flies. After only 5 min, the mutants consumed more food than the controls (*Figure 5C*). The peak levels of ingested food in *trpγ* mutants occurred after 15 min, and the total amount at this time was 1.7 (±0.5)-fold higher than the control flies (*Figure 5C*). After 30 min of feeding, the total amount of internal food declined until 90 min (*Figure 5C*). After this time, the levels reached a steady state and were no longer significantly different than the control flies (*Figure 5C*). The increased levels of ingested food occurred in both sated and

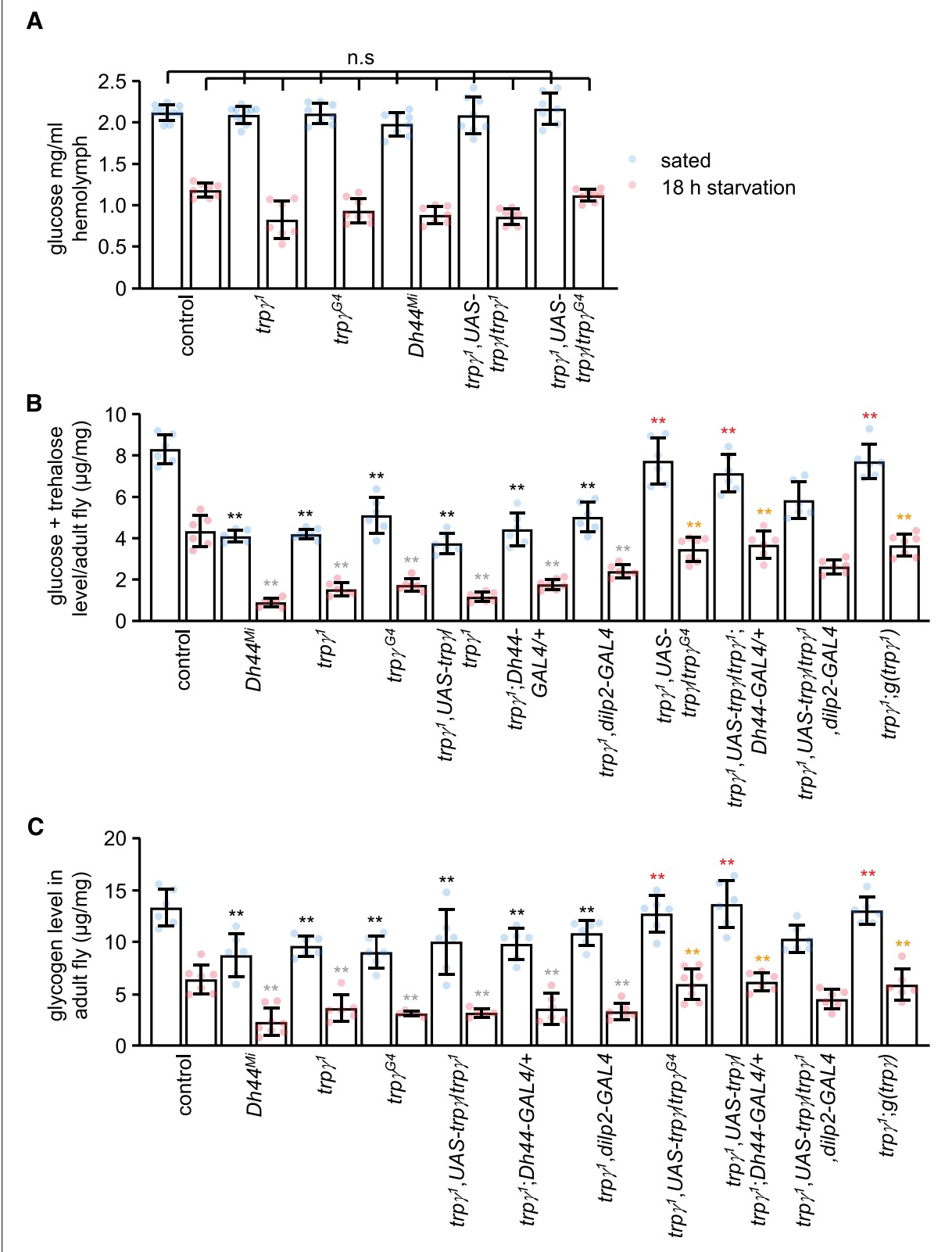

**Figure 4.** Sugars and glycogen levels under sated and starvation conditions. (**A**) Glucose levels in the hemolymph. n = 7–10. (**B**) Trehalose and glucose levels in whole-body extracts. n = 6–7. (**C**) Glycogen levels in whole-body extracts. n = 5–8. Means ± SEMs. Single-factor ANOVA with Scheffe's analysis was used as a post hoc test to compare multiple sets of data. The black and red asterisks indicate significant differences (**p < 0.01) from the sated

*Figure 4 continued on next page*

*Figure 4 continued*

controls and mutants, respectively. The gray and orange asterisks indicate statistical significance (\*\*p < 0.01) from the starved controls and mutants, respectively. n.s., not significant.

The online version of this article includes the following figure supplement(s) for figure 4:

**Figure supplement 1.** Sugars and glycogen levels under fed and starved conditions, and survival of flies under starvation condition.

starved *trpγ* mutant flies (*Figure 5D*). We also observed similar excessive feeding when we inactivated *trpγ* neurons in adults using *UAS-Kir2.1* in combination with the *trpγ-GAL4* and the *tub-GAL80^ts^* (*Figure 5E*). The *trpγ* mutants also consumed a higher level of either D-glucose or L-glucose than control flies (*Figure 5—figure supplement 1*).

To determine whether *trpγ* functions in controlling food ingestion through DH44 neurons, we performed rescue experiments. We found that the phenotype was rescued by expressing *UAS-trpγ* in DH44 neurons (*Figure 5F*). We also rescued the phenotype with the *trpγ* genomic transgene, or with the *trpγ-GAL4* in combination with the *UAS-trpγ* (*Figure 5F*). These data indicate that *trpγ* function is required in DH44 neurons for controlling the level of ingested food. Furthermore, inhibition of DH44 neurons with Kir2.1 caused an increase in consumption (*Figure 5G*). However, *Dh44^Mi^* mutants consumed a similar level of food as the controls (*Figure 5D*, *Figure 5—figure supplement 1*).

## TRPγ is necessary for crop physiology

The increased levels of ingested food combined with expression of *trpγ* in neuronal varicosities lining the crop raise the possibility that *trpγ* might impact crop physiology. To test this idea, we allowed control and mutant male flies to feed on 5% sucrose for 12 or 24 hr, and then dissected the abdomens to monitor the distribution of crop sizes and crop movements, respectively. To analyze the size of the crop, which expands upon food ingestion, we used a scoring system scaled from 1 to 5, with 1 indicating no dye in the crop and 5 for maximum crop distension (*Edgecomb et al., 1994*; *Figure 6A*). Approximately half (55.8% ± 2.1%) of the control flies maintained their crop at an intermediate size of 3, with only 9.7% ± 2.7% displayed a full crop extension (*Figure 6B*). We then tested whether feeding on nutritive D-glucose versus non-nutritive L-glucose altered crop size. We found that 55.4% ± 2.8% of the controls fed on D-glucose had an intermediate size of 3, while only 13.0% ± 1.2% of the crops were completely filled and had a score of 5 (*Figure 6C*). However, if the control flies were offered L-glucose, the proportion of flies that showed an intermediate sized crop declined (38.9% ± 2.2%), and the percentage of filled crops increased (27.4% ± 1.9%; *Figure 5C*). Thus, crop size appeared to be impacted by nutritional value. Consistent with this observation, the distribution of crop sizes with other nutritive sugars was similar to nutritive D-glucose (*Figure 6—figure supplement 1A–D*).

We found that crop sizes were significantly altered in the *trpγ* mutant. If the animals were presented sucrose, the percentage of *trpγ* mutant flies with an intermediate size (34.7% ± 2.0%) was reduced significantly relative to the controls, while the proportion of animals with full crop extension was more than twice as high (28.9% ± 2.7%; *Figure 5B*). We obtained similar results with other nutritious sugars (*Figure 6—figure supplement 1A–D*). When offered D-glucose or L-glucose, the proportion of intermediate filled *trpγ^1^* crops was also lower than control animals, while the percentage of crops completely filled with food (score of 5) was significantly higher than control animals (*Figure 6C*). Moreover, the *trpγ^1^* flies showed similar distributions of crop sizes regardless of whether they were fed on D-glucose or L-glucose (*Figure 6C*). This latter result supports our findings that *trpγ* mutants are compromised in their capacity to differentiate between metabolizable versus non-metabolizable sugar. The *Dh44^Mi^* flies also showed a shift toward the filled crops (5 score) when they consumed sucrose (*Figure 6—figure supplement 1E*). The differences in crop sizes when *Dh44^Mi^* flies were fed D-glucose or L-glucose were relatively minor, except for higher percentages of crops with a score of 4 when fed on L-glucose (*Figure 6—figure supplement 1E*). The *trpγ^1^* and *Dh44^Mi^* flies also displayed similar defects when fed on D-trehalose, D-maltose, and D-fructose (*Figure 6—figure supplement 1B–D*). The *trpγ* mutant phenotype was rescued by the expression of *UAS-trpγ* under control of the *trpγ-GAL4* (*trpγ^G4^*) or the *Dh44-GAL4*, but not the *dilp2-GAL4* (*Figure 6B*).

Crop contractions affect ingestion and the size of the meals. Therefore, we wondered whether the frequency of crop contractions was altered by the *trpγ* mutation. To address this question, we fed control and *trpγ* mutants 5% sucrose, and then quantified the number of contractions per minute.

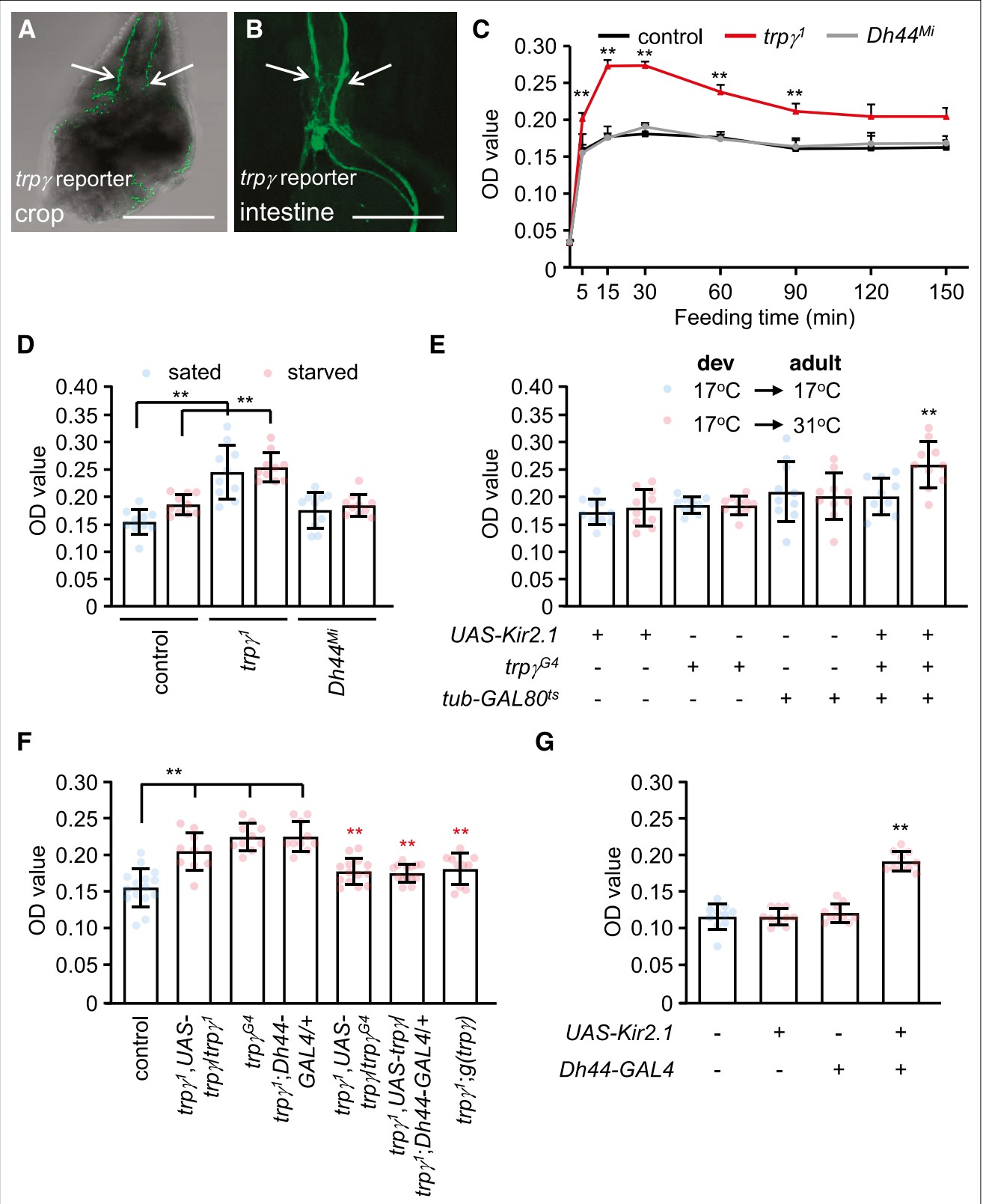

**Figure 5.** Requirement for *trpγ* in diuretic hormone 44 (DH44) neurons to control consumption. (**A–B**) Confocal images of *trpγ* reporter expression (*trpγ^{G4}/UAS-mCD8::GFP*) in (**A**) the crop and (**B**) the intestine after staining the tissues with anti-GFP. Scale bars represent 25 μm. (**C–G**) Quantification of the levels of ingested food was determined by feeding flies 1% agarose containing 5% sucrose and blue food dye. The concentration of food was measured by determining the ODs (629 nm) of extracts from whole male flies. (**C**) The indicated sated flies were allowed to feed for various periods

*Figure 5 continued on next page*

*Figure 5 continued*

of time prior to quantification of internalized food. n = 10–12. (**D**) Levels of internalized food in the indicated sated or starved flies after feeding for 30 min. n = 10. (**E**) Feeding levels after inactivating *trpγ* neurons in adults only with *Kir2.1*. The GAL4 repressor (GAL80$^{ts}$) is active at 17°C and inactive at 31°C. The flies were for sated for 30min before intiating the feeding assays. "Dev" indicates the temperature during all developmental stages through eclosion. n = 10. (**F**) Rescue of the *trpγ* phenotype (high levels of internal food) with the *UAS-trpγ* and the *Dh44-GAL4*, or with the *trpγ* genomic transgene. Red asterisks indicate significant differences from the *trpγ¹* mutant. n = 10–15. (**G**) Measurement of internalized food after silencing *Dh44* neurons with *Kir2.1*. The flies were sated flies for 30 min prior to initiating the feeding assays. n = 10. Means ± SEMs. Single-factor ANOVA with Scheffe's analyses was used as a post hoc test to compare multiple sets of data. The black and red asterisks indicate significance differences from the controls and mutants, respectively (**p < 0.01).

The online version of this article includes the following figure supplement(s) for figure 5:

**Figure supplement 1.** Internal food consumption.

Control animals produce 20.0 ± 0.6 contractions/min (***Figure 6D***), and this frequency was similar with each of the nutritious sugars tested (***Figure 6E***). Moreover, the rate of contractions was highest in crops with an intermediate score of 3 (***Figure 6F***).

We found that the rate of contraction in the *trpγ* mutant was increased significantly (***Figure 6D***). The elevated contraction rate occurred regardless of the nutritive sugar tested (***Figure 6E***). More-over, unlike control flies, in which the contraction frequency was lower with the non-metabolizable L-glucose (***Figure 6G***), the *trpγ¹* showed higher crop contractions after consuming either D-glucose or L-glucose (***Figure 6G***). Thus, the crop contained more food despite having a higher contraction rate. The *Drosophila drop-dead* mutant exhibits a similar phenotype, which has been proposed to be due to a regulatory impairment that reduces entry of food into the midgut (***Peller et al., 2009***). We suggest a similar explanation for the *trpγ* mutant animals. However, the contraction rates were similar between control and *Dh44$^{Mi}$* flies (***Figure 6D and G***). We restored normal crop rates in *trpγ* mutants with the wild-type genomic transgene or by driving *UAS-trpγ* expression with the *trpγ-GAL4*, but not with the *Dh44-GAL4* (***Figure 6D***). These data indicate that Dh44$^+$ neurons contribute to crop size control, but not crop contraction.

## TRPγ regulates defecation

Despite the higher levels of food ingested by *trpγ¹* flies, there were no significant differences in weight between *trpγ* mutants and control flies under sated or starved conditions (***Figure 7—figure supplement 1***). The *Dh44$^{Mi}$* flies were also similar in weight (***Figure 7—figure supplement 1***). A potential explanation for the greater internal food levels, but lack of weight difference became evident by examining the food content after 15 min. In controls, the total level of food in the flies remains nearly constant subsequent to the first 15 min of feeding (***Figure 5C***). However, in the *trpγ* mutants, the internal level of food declines. By 90 min after initiation of the feeding assay, the amount in the control and *trpγ* flies were similar (***Figure 5C***).

To account for the much greater reduction in internal food content in *trpγ* mutant flies, we performed defecation assays. We allowed male flies to feed on 5% sucrose mixed with blue food dye for 24 hr to saturate all the excreta with dye. We then transferred the flies to empty vials and tabulated the excreta over time. We found that the cumulative number of excreta was greater in *trpγ* mutant flies relative to controls (***Figure 7A and B***). Conversely, as previously demonstrated, *Dh44* mutants produced significantly less excreta than controls (***Dus et al., 2015***; ***Figure 7A and B***). In addition to an increase in the number of defecation events, the size of the excreta was increased in the *trpγ* mutants (***Figure 7C***). Consistent with the increased number of events and size of the excreta, the total level of defecation was increased (***Figure 7D***). The excreta were also larger in the *Dh44* mutants (***Figure 7C***), but total defecation was not different from the controls (***Figure 7D***), due to the fewer number of excreta produced.

To determine the cells that require *trpγ* function for normal defecation, we conducted rescue experiments. We found that the increased defecation phenotype exhibited by the *trpγ* mutants was rescued with the genomic *trpγ* transgene, and with *UAS-trpγ* driven by the *trpγ-GAL4* (*trpγ$^{G4}$*), but not with the *Dh44-GAL4* or the *dilp2-GAL4* (***Figure 7C and D***). This indicates that the increased defeca-tion in *trpγ* mutants is mediated by cells other than DH44 and dILP2 neuroendocrine cells. While the relevant neurons remain to be identified, neurons have been described that impact on defecation, some of which are in the central nervous system (***Cognigni et al., 2011***).

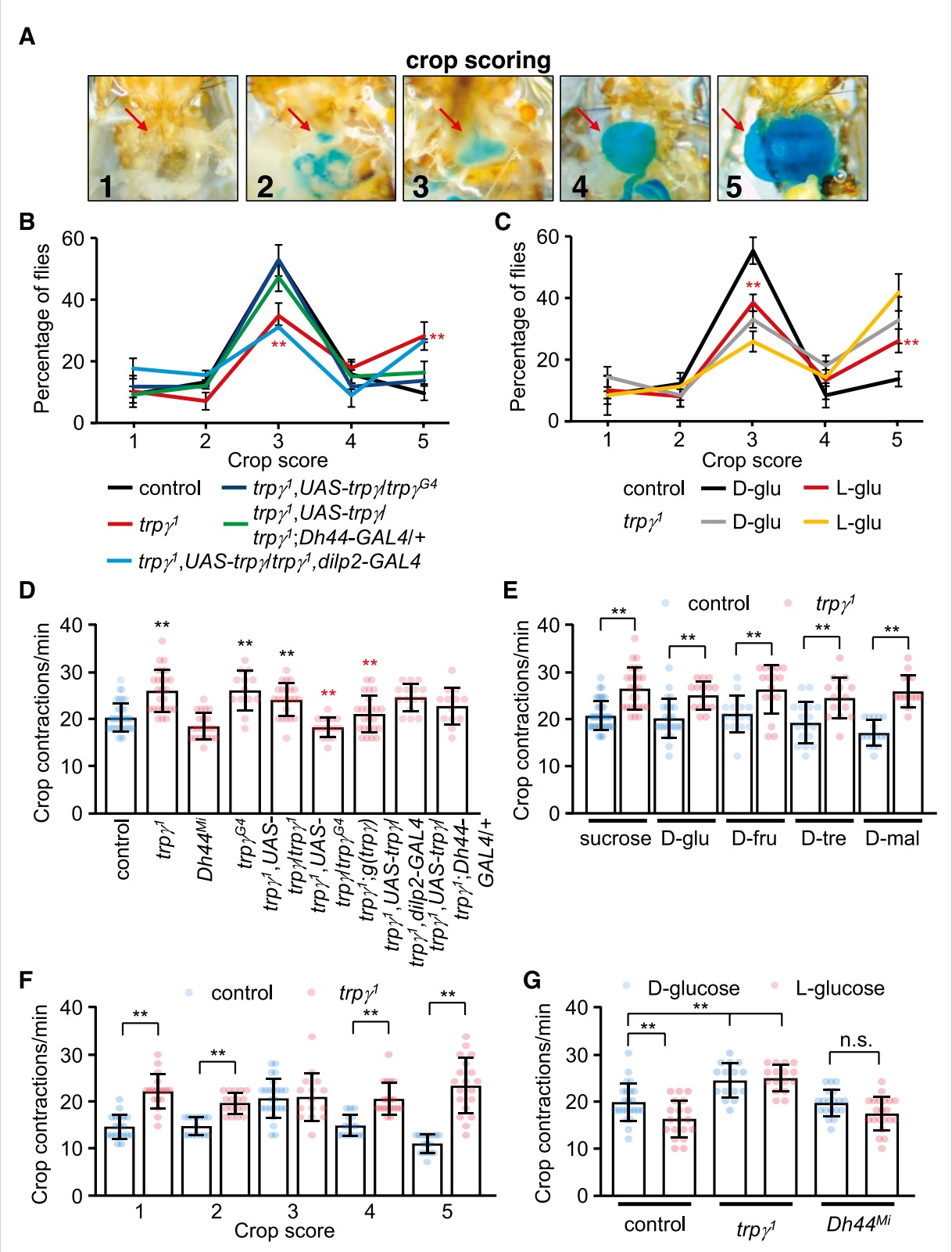

**Figure 6.** Crop extension and contraction rates. (**A**) Representative images of crop sizes scored on a 1–5 scale after feeding for 12 hr 1% agarose containing 5% sucrose and blue food coloring. The images were acquired with an *i*solution IMTcam3 camera connected to a Nikon microscope. (**B**) The percentages of flies with crop sizes on a 1–5 scale after feeding for 12 hr on 5% sucrose. n = 5 (16–20 flies were used for each experiment: total 92–100 flies). Each genotype was compared with the control at each 1–5 scale. (**C**) The percentages of flies with crop sizes on a 1–5 scale after feeding on 5%

*Figure 6 continued on next page*

*Figure 6 continued*

D-glucose (D-glu) or 5% L-glucose (L-glu). n = 5 (15–20 flies were used for each experiment: total 75–80 flies). Comparisons were performed between flies of the same genotype that fed on either D-glu or L-glu. (**D**) Crop contraction rates after feeding on 5% sucrose (includes all scales). n = 13–34. (**E**) Crop contraction rates as a function of crop scale (1–5) after feeding on 5% sucrose. n = 14–31. (**F**) Crop contraction rates (all crop scales combined) after feeding on 5% D-glucose or 5% L-glucose. n = 14–27. (**G**) Crop contraction rates (all crop scales) after feeding on 5% sucrose, D-glu, D-fructose (D-fru), D-trehalose (D-tre), or D-maltose (D-mal). n = 14–23. Means ± SEMs. Single-factor ANOVA with Scheffe's analysis was used as a post hoc test to compare multiple sets of data. The data in (**F**) were analyzed using unpaired Student's t-tests. The red asterisks in (**B**) indicate significance from controls. The red asterisks in (**C**) indicate significance within same genotype for different sugars. n.s., not significant. (**p < 0.01).

The online version of this article includes the following figure supplement(s) for figure 6:

**Figure supplement 1.** Scoring crop sizes after feeding on different sugars.

## Discussion

### TRPγ is required in the brain for post-ingestive selection of nutritive sugar

In this study, we reveal that expression of TRPγ in the brain is required for the starvation-induced switch in preference from the sweeter but non-metabolizable L-glucose to the less sweet but nutritive D-glucose. The strong shift in bias to D-glucose due to caloric deprivation represents a post-ingestive behavior, as it occurs in the absence of peripheral sugar receptors (*Dus et al., 2015*). This selection of nutritive sugar in starved animals depends on DH44 neuroendocrine cells in the PI region of the brain (*Dus et al., 2015*). In support of our conclusion that *trpγ* functions in these neuroendocrine cells, we rescued the mutant phenotype using the *Dh44-GAL4* in combination with *UAS-trpγ*, and we replicated the mutant phenotype by RNAi-mediated knockdown of *trpγ* with the same driver. While we demonstrate that TRPγ is required in DH44 neurons in the brain, TRPγ is widely expressed. Therefore, we do not exclude that the contribution of TRPγ to the regulation of feeding behavior is influenced by expression in other neurons, including sensory neurons.

### TRPγ and potential basis for attenuation of neuronal firing

Mutation of a TRP channel would be expected to decrease excitation of neurons. Surprisingly, mutation of *trpγ* caused the opposite effect, as we found significantly higher peak signals using the genetically encoded voltage indicator, ASAP2s. This greater response is indicative of more rapid firing of action potentials (*Chamberland et al., 2017*). We suggest that the underlying basis for this effect may be due to a role for TRPγ that is similar to a function of TRPC4, which is a mammalian TRPC homolog in lateral septal neurons (*Tian et al., 2014*). Activation of TRPC4 has been shown to cause two types of responses, one of which is 'plateau depolarization'—a period characterized by few if any action potentials due to strong depolarization, thereby inactivating voltage-gated $Na^+$ channels (*Tian et al., 2014*). Moreover, the plateau depolarization caused by TRPC4 is dependent on activation of the channel by $Ca^{2+}$ influx (*Tian et al., 2014*).

We propose that similar to TRPC4 in lateral septal neurons (*Tian et al., 2014*), TRPγ functions in DH44 neurons to hold these neurons in an afterdepolarization state, reducing the rate of firing. We suggest that in *trpγ* mutant DH44 neurons, the loss of the afterdepolarization block underlies the higher peak ASAP2s signal (increased firing rate) and higher baseline activity. We also propose that the TRPγ-dependent afterdepolarization requires extracellular $Ca^{2+}$, as is the case for TRPC4. Consistent with this idea, when we measured the ASAP2s signals in control DH44 neurons bathed in a $Ca^{2+}$-free buffer, the cells displayed increased ASAP2s fluorescence, similar to the *trpγ* mutants. At the physiological level, this would result in the release of excessive DH44, and deplete the neuropeptide stores, thereby rendering them unable to respond to a continuous elevation of nutrients in the hemolymph.

### Metabolic defect due to loss of TRPγ decreases survival during starvation

To reveal whether the impairment in starvation-induced D-glucose selection is associated with a metabolic defect, we examined circulating sugar levels. While there were small reductions in the starved mutants, the differences with the control animals were not significant. In contrast, there were large and highly significant decreases in intracellular sugar concentrations in both sated and starved flies.

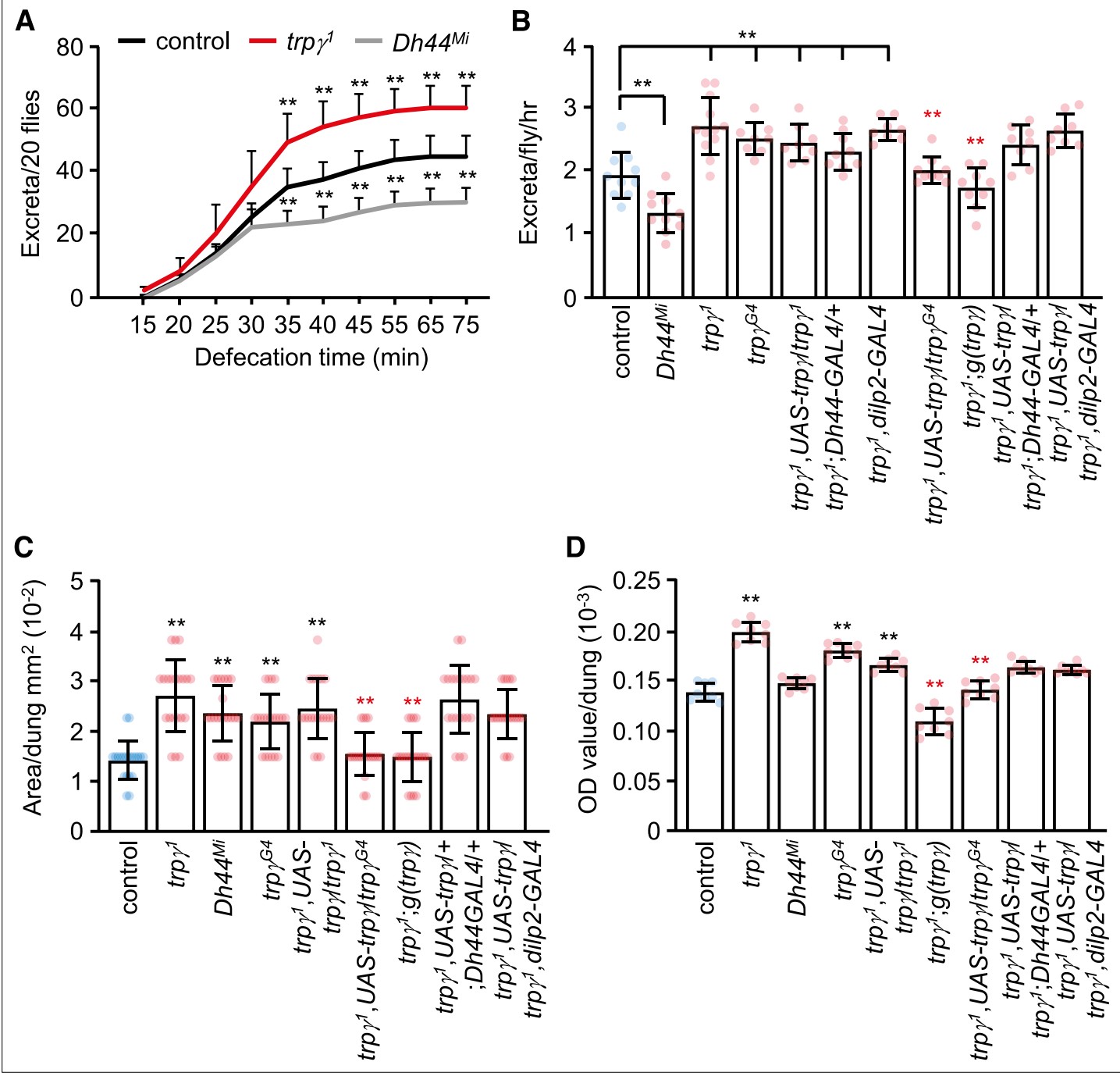

**Figure 7.** Rates and quantities of defecation. (**A**) Cumulative defecation rates of male flies. n = 9. (**B**) Defecation rates of flies after 1 hr. n = 8–12. (**C**) Quantification of excreta size (area) of indicated male flies. n = 20. (**D**) Quantification of defecation amount per dung with the indicated male flies. n = 8. Means ± SEMs. Single-factor ANOVA with Scheffe's analysis was used as a post hoc test to compare multiple sets of data. The black and red asterisks indicate significance from the controls and mutants, respectively (**p < 0.01).

The online version of this article includes the following figure supplement(s) for figure 7:

**Figure supplement 1.** Body weight measurements.

We suggest that a key deficit in *trpγ* mutants is glucose uptake. However, glucose uptake has not been assayed in flies, and we were also unable to develop such measurements. Nevertheless, in support of the concept that there is a defect in glucose uptake, we found that the glycogen stores were also diminished in *trpγ* mutants. In mammals, there is evidence that taste-independent sugar preference is

influenced by glucose metabolism (*Ren et al., 2010*). Thus, it is intriguing to speculate that the deficit in D-glucose selection in *trpγ* mutants is due in part to impairment in metabolism of D-glucose.

Our findings that *trpγ* function is rescued by expressing the wild-type transgene in DH44 neurons, and *Dh44* mutant flies exhibited the same deficits in tissue sugar and glycogen levels raise the possibility that TRPγ promotes the release of DH44, which in turn increases glucose uptake. Consistent with this proposal, corticotropin-releasing hormone, which is the mammalian homolog of DH44, increases glucose uptake in a variety of cell types (*Hogg et al., 2018*; *Lu et al., 2018*). We suggest that TRPγ is critical for establishing the glycogen stores that enable the animals to survive under starvation conditions. In support of this latter idea, survival of the *trpγ* mutant flies under starvation conditions is significantly shorter than in control animals.

### Broad requirements for TRPγ in responding to starvation

Throughout the animal kingdom, the brain is essential for sensing the internal energy state, and then for promoting homeostatic physiological and behavioral changes, such as the regulation of feeding levels. Consistent with the metabolic deficits exhibited by the *trpγ* mutants, such as low glycogen stores, we found that the flies exhibited an increase in feeding. However, they did not gain weight, and this was presumably due to an increase in defecation, which offsets the higher level of ingestion. The crop is the fly equivalent of the mammalian stomach, and we found that in the *trpγ* mutants there was a significantly higher proportion of filled crops, and an elevation in crop contractions. Similar to the requirements for TRPγ in DH44 neurons for post-ingestive selection of D-glucose, and for establishing normal glycogen concentrations, we found that feeding levels and crop size also depended on TRPγ in DH44 neurons. The *trpγ* and *Dh44* mutants displayed similar deficits in starvation-induced selection of D-glucose, intracellular sugar and glycogen levels, and increased crop filling. However, the *Dh44* mutants did not show an increase in feeding. Therefore, although TRPγ functioned in DH44 neurons, the roles of DH44 and TRPγ were not identical.

While multiple TRPγ functions depend on expression in DH44 neurons, the crop contraction levels, and the size and number of excretions reflect roles for TRPγ in other cells. Consistent with these findings, the *Dh44^Mi* mutant exhibits normal crop contractions. Moreover, the total defecation level was unchanged, although this is a consequence of few excreta of larger size. TRPγ is widely expressed in the brain, and the neurons requiring TRPγ for these latter functions remain to be determined.

### Potential roles for mammalian TRPs in neuroendocrine function

Our work establishes a diversity of post-ingestive roles of the TRPγ channel in the homeostatic control of metabolism, which largely depend on TRPγ expression in the brain. It is notable that many TRP channels are expressed in the mammalian brain, including neuroendocrine cells (*Dhakal and Lee, 2019*; *Gao et al., 2017*; *Gavello et al., 2016*; *Islam, 2020*; *Kumar et al., 2017*; *Leinders-Zufall and Boehm, 2014*). However, their roles in regulating metabolism through the brain-gut axis are poorly understood, but have been the subject of wide speculation (*Dhakal and Lee, 2019*; *Gavello et al., 2016*; *Islam, 2020*; *Leinders-Zufall and Boehm, 2014*). Similar to the requirement for TRPγ in Dh44 neuroendocrine cells, which control taste-independent feeding in flies, we suggest that TRP channels in neuroendocrine cells in the mammalian brain might also function in the regulation of taste-independent food selection, and in homeostatic control of metabolism, in response to the internal state.

## Materials and methods

**Key resources table**

| Reagent type (species) or resource | Designation | Source or reference | Identifiers | Additional information |
|---|---|---|---|---|
| Genetic reagent (*Drosophila melanogaster*) | *trpγ¹* | *Akitake et al., 2015* | | |
| Genetic reagent (*Drosophila melanogaster*) | *trpγ^G4* | *Akitake et al., 2015* | | |

*Continued on next page*

*Continued*

| Reagent type (species) or resource | Designation | Source or reference | Identifiers | Additional information |
|---|---|---|---|---|
| Genetic reagent (*Drosophila melanogaster*) | *trpγ¹,UAS- trpγ*/CyO | *Akitake et al., 2015* | | |
| Genetic reagent (*Drosophila melanogaster*) | *trpγ¹;g(trpγ)* | *Akitake et al., 2015* | | |
| Genetic reagent (*Drosophila melanogaster*) | *Dh44^Mi* | Bloomington *Drosophila* Stock Center | stock # 1454; RRID:BDSC_1454 | Provided by Y Kim (GIST) and Dr GS Suh (KAIST) |
| Genetic reagent (*Drosophila melanogaster*) | *Dh44-GAL4* | Bloomington *Drosophila* Stock Center | stock # 1453; RRID:BDSC_1453 | Provided by Y Kim (GIST) and Dr GS Suh (KAIST) |
| Genetic reagent (*Drosophila melanogaster*) | *UAS-hid*/CyO | Bloomington *Drosophila* Stock Center | stock # 65403; RRID:BDSC_65403 | |
| Genetic reagent (*Drosophila melanogaster*) | *UAS-mCD8::GFP* | Bloomington *Drosophila* Stock Center | stock # 5130; RRID:BDSC_5130 | |
| Genetic reagent (*Drosophila melanogaster*) | *UAS-nls::tdtomato* | *Knapp et al., 2015* | | Provided by S Marella |
| Genetic reagent (*Drosophila melanogaster*) | *UAS-dicer2;UAS-trpγRNAi* | *Akitake et al., 2015* | Stock # 107656 VDRC-107656 | |
| Genetic reagent (*Drosophila melanogaster*) | *iav-GAL4* | *Kwon et al., 2010* | | |
| Genetic reagent (*Drosophila melanogaster*) | *dilp2-GAL4* | *Rulifson et al., 2002* | | Provided by EJ Rulifson |
| Genetic reagent (*Drosophila melanogaster*) | *Gr64^ab* | *Jiao et al., 2007* | | |
| Genetic reagent (*Drosophila melanogaster*) | *trpA1¹* | *Kwon et al., 2008* | | |
| Genetic reagent (*Drosophila melanogaster*) | *amo¹* | *Watnick et al., 2003* | | |
| Genetic reagent (*Drosophila melanogaster*) | *trpml²* | *Venkatachalam et al., 2008* | | |
| Genetic reagent (*Drosophila melanogaster*) | *trp³⁴³* | *Wang et al., 2005* | | Originally isolated from W Pak lab |
| Genetic reagent (*Drosophila melanogaster*) | *pyx³* | *Lee et al., 2005* | | |
| Genetic reagent (*Drosophila melanogaster*) | *wtrw^ex* | *Wang et al., 2005* | | |
| Genetic reagent (*Drosophila melanogaster*) | *trpl^29134* | *Niemeyer et al., 1996* | | |
| Genetic reagent (*Drosophila melanogaster*) | *pain²* | *Tracey et al., 2003* | | |
| Genetic reagent (*Drosophila melanogaster*) | *nan^36a* | *Kim et al., 2003* | | |

*Continued on next page*

*Continued*

| Reagent type (species) or resource | Designation | Source or reference | Identifiers | Additional information |
|---|---|---|---|---|
| Genetic reagent (*Drosophila melanogaster*) | *UAS-GCaMP6s* | Bloomington *Drosophila* Stock Center | stock # 77131; RRID:BDSC_77131 | |
| Genetic reagent (*Drosophila melanogaster*) | *UAS-dsRed* | Bloomington *Drosophila* Stock Center | stock # 59853; RRID:BDSC_59853 | |
| Genetic reagent (*Drosophila melanogaster*) | *20xUAS-ASAP2s* | Bloomington *Drosophila* Stock Center | stock# 76247; RRID:BDSC_76247 | |
| Genetic reagent (*Drosophila melanogaster*) | *UAS-Kir2.1* | Bloomington *Drosophila* Stock Center | stock # 6596; RRID:BDSC_6596 | |
| Genetic reagent (*Drosophila melanogaster*) | *iav$^{3621}$* | Bloomington *Drosophila* Stock Center | stock # 24768; RRID:BDSC_24768 | Generated by Dr P Salvaterra |
| Genetic reagent (*Drosophila melanogaster*) | *tubulin-GAL80$^{ts}$* | Bloomington *Drosophila* Stock Center | stock # 7017; RRID:BDSC_7017 | |
| Antibody | Mouse anti-GFP (mouse monoclonal) | Molecular Probes | A-11120 | (1:1000, 2 µL) |
| Antibody | Rabbit anti-DsRed (rabbit polyclonal) | Clontech | 632496 | (1:1000, 2 µL) |
| Antibody | Goat anti-mouse IgG (mouse polyclonal) | Thermo Fisher/Invitrogen | A-11031 | Alexa Fluor 488 (1:200, 10 µL) |
| Antibody | Goat anti-rabbit IgG (rabbit polyclonal) | Thermo Fisher/Invitrogen | A-11034 | Alexa Fluor 568 (1:200, 20 µL) |
| Antibody | Rabbit anti-DH44 (rabbit polyclonal) | *Cabrero et al., 2002* | | (1:500, 15 µL) |
| Commercial assay or kit | Glucose (HK) Assay Kit | Sigma-Aldrich | Sigma: G3293 | |
| Chemical compound, drug | Brilliant blue FCF | Takeda-Wako Pure Chemical Industry Ltd | Cat. 3844-45-9 | |
| Chemical compound, drug | Paraformaldehyde | Electron Microscopy Sciences | Cat. 15710 | |
| Chemical compound, drug | Sulforhodamine B | Sigma-Aldrich | Cat. 3520-42-1 | |
| Chemical compound, drug | Sucrose | Sigma-Aldrich | Cat. 57-50-1 | |
| Chemical compound, drug | D-(+)-Glucose | Sigma-Aldrich | Cat. 50-99-7 | |
| Chemical compound, drug | D-(-)-Fructose | Sigma-Aldrich | Cat. 57-48-7 | |
| Chemical compound, drug | D-(+)-Maltose | Sigma-Aldrich | Cat. 6363-53-7 | |
| Chemical compound, drug | D-(+)-Trehalose | Sigma-Aldrich | Cat. 6138-23-4 | |
| Chemical compound, drug | KCl | Sigma-Aldrich | Cat. 7447-40-7 | |
| Chemical compound, drug | Hydrochloric acid | Samchun | Cat. 7647-01-0 | |
| Chemical compound, drug | Ethanol | Merck | Cat. 64-17-5 | |
| Chemical compound, drug | Trehalase from porcine kidney | Sigma-Aldrich | Cat. 9025-52-9 | |
| Chemical compound, drug | Amyloglucosidase from *Aspergillus* | Sigma-Aldrich | Cat. 9032-08-0 | |

## Fly strains and husbandry

All flies were maintained at 25°C (unless otherwise indicated) under 12 hr light/12 hr dark cycles. The control strain used in this study was *w1118*. We previously described *trpγG4* (*Akitake et al., 2015*), *trpγ1* (*Akitake et al., 2015*), *trpγ1,UAS-trpγ/CyO* (*Akitake et al., 2015*), *trpγ1;g(trpγ)* (*Akitake et al., 2015*), *Gr64ab* (*Jiao et al., 2007*), *trpA11* (*Kwon et al., 2008*), *amo1* (*Watnick et al., 2003*), *trpml2* (*Venkatachalam et al., 2008*), *trp343* (*Wang et al., 2005*) (originally isolated from W Pak lab), *pyx3* (*Lee et al., 2005*) and *wtrwex* (*Wang et al., 2005*). The following lines were described previously by other groups: *trpl29134* (*Niemeyer et al., 1996*), *pain2* (*Tracey et al., 2003*), *nan36a* (*Kim et al., 2003*), and *dilp2-GAL4* (*Rulifson et al., 2002*) (provided by EJ Rulifson). *UAS-trpγRNAi* (transformant ID 107656) was from the Vienna *Drosophila* Research Center. *Dh44Mi* (*Dus et al., 2015*) and *Dh44-GAL4* (*Dus et al., 2015*) were provided by Y Kim (GIST) and Dr GS Suh (KAIST). *UAS-nls::tdtomato* (*Knapp et al., 2015*) was provided by S Marella. We obtained the following lines from the Bloomington *Drosophila* Stock Center (https://bdsc.indiana.edu/): *iav-GAL4* (*Kwon et al., 2010*), *UAS-GCaMP6s* (#77131), *UAS-mCD8::GFP* (#5130), *UAS-dsRed* (#59853), *UAS-Kir2.1* (#6596), *iav3621* (generated by Dr P Salvaterra), and *tubulin-GAL80ts* (#7017). Additional information regarding specific stains and genotypes is provided in the Materials and methods section.

## Chemical sources

The following chemicals and kit were purchased from Sigma-Aldrich Co.: sulforhodamine B (cat. # 3520-42-1), sucrose (cat. # 57-50-1), D-glucose (cat. # 50-99-7), L-glucose (cat. # 921-60-8), D-fructose (cat. # 57-48-7), D-maltose (cat. # 6363-53-7), D-trehalose (cat. # 6138-23-4), trehalase from porcine kidney (cat. # 9025-52-9), amyloglucosidase (cat. # 9032-08-0), and the glucose (HK) Assay Kit (Sigma: G3293). Brilliant blue FCF (cat. # 3844-45-9) was purchased from Wako Pure Chemical Industry Ltd.

## Binary food choice assays

We performed binary food choice assays as previously described (*Dus et al., 2013*; *Lee et al., 2009*). Prior to starvation, the flies were maintained on standard cornmeal/molasses fly food. To conduct the assays, 40–50 flies (3–6 days of age; mixed sexes, unless indicated otherwise) were starved for 5 hr (sated) or 18 hr (starved) in a humidified chamber. Two different food sources were prepared containing 1% agarose: one containing 50 mM D-glucose, and the other containing 200 mM L-glucose. These food sources were mixed with either blue food coloring (brilliant blue FCF, 0.125 mg/mL) or red food coloring (sulforhodamine B, 0.1 mg/mL). The two mixtures were distributed in a 72-well microtiter dish in alternating wells (Thermo Fisher Scientific, cat. # 438733). We introduced the starved flies into the dish, kept them in a dark, humidified chamber, and allowed the flies to feed for 90 min at 25°C. Then, we sacrificed the flies by putting them in a freezer and analyzed the color of their abdomens under a stereomicroscope. Blue ($N_B$), red ($N_R$), or purple ($N_P$) flies were tabulated. Each preference index (PI) was calculated according to the following equation: $(N_B-N_R)/(N_R +N_B + N_P)$ or $(N_R-N_B)/(N_R +N_B + N_P)$, depending on the dye/tastant combinations. PIs = 1.0 and -1.0 indicate complete preference for one over the other food. A PI = 0 indicates no bias between the two food choices.

## Immunohistochemistry

We performed immunohistochemistry as previously described (*Lee and Montell, 2013*). Briefly, to fix and block the specimens, we placed freshly dissected tissues into a well of a 24-well Tissue Culture plate (Costar Corp.) placed on ice, which contained 940 µL of Fix Buffer (0.1 M Pipes pH 6.9, 1 mM EGTA, 1% Triton X-100, 2 mM $MgSO_4$, 150 mM NaCl) and 60 µL of 37% formaldehyde. Formaldehyde was added to the wells containing the Fix buffer and mixed immediately before adding the tissue. We transferred as many dissected tissues into the Fix buffer as we could dissect in 15 min. The tissues were incubated for another 30 min, washed with Wash buffer (1× PBS, 0.2% saponin), and blocked for 4–8 hr at 4°C with 1 mL of Blocking buffer (1× PBS, 0.1% saponin, and 5 mg/mL BSA).

To perform immunostaining, the tissues were transferred and incubated overnight at 4°C with the primary antibodies in the Blocking buffer, washed three times with Wash buffer for 15 min each on ice, incubated with the secondary antibodies (1:200) for 4 hr at 4°C, and washed three times with Wash buffer for 15 min each on ice. The antibodies were used at the following dilutions: mouse anti-GFP (1:1000, Molecular Probes, cat. # A11120), rabbit anti-DsRed (Clontech, cat. # 632496), rabbit anti-DH44 (1:500) (*Cabrero et al., 2002*), goat anti-mouse (Alexa488 cat. # A11029), and goat anti-rabbit

Alexa568 (cat. # A11036). The tissues were transferred into 1.25× PDA Dilution buffer (37.5% glycerol, 187.5 mM NaCl, 62.5 mM Tris pH 8.8), and incubated >1 hr at 4°C. Samples were mounted and viewed using a Carl Zeiss LSM 700 confocal microscope.

We performed staining to quantify cellular DH44 neuropeptide secretion as previously described (*Dus et al., 2015*). Briefly, we dissected brains from 18 hr-starved male flies in sugar-free *Drosophila* saline (108 mM NaCl, 5 mM KCl, 8.2 mM $MgCl_2$, 2 mM $CaCl_2$, 4 mM $NaHCO_3$, 1 mM $NaH_2PO_4$, 5 mM HEPES pH 7.5), and immediately transferred the brains to *Drosophila* saline containing the indicated concentrations of D-glucose, L-glucose, or D-fructose for 30 min at room temperature. Staining with anti-DH44 was conducted as described above. Image acquisition was performed with a Leica Stellaris 5 Confocal Microscope. The ImageJ (Fiji) application was used to quantify the fluorescence intensity of the intracellular DH44 neuropeptide stained with anti-DH44 antibody in the brain.

## Ex vivo Ca²⁺ imaging

Ex vivo $Ca^{2+}$ imaging was performed as previously described (*Inagaki et al., 2014*). Briefly we dissected adult brains from 7- to 10-day-old males in cold sugar-free *Drosophila* imaging saline (108 mM NaCl, 5 mM KCl, 8.2 mM $MgCl_2$, 2 mM $CaCl_2$, 4 mM $NaHCO_3$, 1 mM $NaH_2PO_4$, 5 mM HEPES pH 7.5) and immobilized the brains with a slice harp (SHD-26GH/10, Warner Instruments) on a 35 mm plastic Petri dish (35 3001 Falcon), containing 4 mL sugar-free *Drosophila* imaging saline. We imaged the $Ca^{2+}$ dynamics using a Zeiss LSM 700 confocal microscope with a Zeiss 20× water objective (20×/1.0 DIC (uv) VIS-IR, Zeiss) and a 488 nm laser. We recorded each brain for 20 min in total (512 × 512 pixels) at a rate of 400 ms/frame. Approximately 15 Z axial sections were imaged in one time-series cycle. The section interval was ~6 μm. The time intervals between each cycle were ~8 s. Before stimulating a brain, we imaged the basal GCaMP6s signals for 5 min. Pseudo-color images and image analyses were performed using ImageJ. ΔF/F (%) was calculated as $(F_{max}-F_0)/F_0 \times 100\%$. $F_0$ is the GCaMP6s baseline value averaged for 10 frames immediately before glucose application. $F_{max}$ is the maximum fluorescence value following glucose delivery.

## Ex vivo voltage imaging

To perform the ex vivo voltage imaging, we generated flies that express the voltage sensor, ASAP2s (*Chamberland et al., 2017*), in DH44 neurons. Briefly, we dissected brains from 7- to 10-day-old males in cold sugar-free *Drosophila* imaging saline (108 mM NaCl, 5 mM KCl, 8.2 mM $MgCl_2$, 2 mM $CaCl_2$, 4 mM $NaHCO_3$, 1 mM $NaH_2PO_4$, 5 mM HEPES pH 7.5). For the no-$Ca^{2+}$ experiments, we prepared $Ca^{2+}$-free imaging saline (108 mM NaCl, 5 mM KCl, 8.2 mM $MgCl_2$, 4 mM $NaHCO_3$, 1 mM $NaH_2PO_4$, 5 mM HEPES, 0.5 mM EGTA pH 7.5). We transferred the brains to glass coverslips coated with poly-D-lysine (VWR BD354086) surrounded by silicone to keep the *Drosophila* imaging saline on the microscope slides. After placing the slides on the microscope stage, we perfused the *Drosophila* imaging saline from one side (delivered via gravity) and adjusted the vacuum on the opposite side to match the flow rate of the perfusion. The buffer delivery system was prepared by connecting standard 1 mm diameter tubing to 50 mL syringes placed higher than the imaging area. We connected the tubing to the syringes using luer valves, and controlled the buffer delivery with roller clamps (World Precision Instruments, 14011). We connected the other end of the tubing to a stainless steel needle (45° angle, 0.010" inner diameter, McMaster-Carr, 75165 A62) to deliver solutions to the brain samples. The perfusion system allowed for switching among three different solutions in one experiment.

We imaged the ASAP2s signals using a Bruker two-photon microscope with a Mai Tai DeepSee laser set at 920 nm (Mai Tai EHP 1040 DS) and an Olympus 40× water objective (LUMPlanFL N 40×/0.80W) in the Galvo scanning mode. We selected a single neuron as the region of interest, and adjusted the laser strength and PMT gain. We recorded each brain for 6 min in total (512 × 512 pixels) at a rate of ~35 ms/frame. We imaged a single axial section using a one time-series cycle. We stimulated the brains by switching to the various buffers using roller clamps to turn the buffer flow on and off. Before stimulating a brain, we imaged the basal ASAP2s signals for 60 s. We then started perfusing the brain with 20 mM D-glucose, L-glucose, or buffer as a control for 120 s. We then switched back to the imaging saline. Brightness over time data were collected in real time using Prairie View software (Bruker Prairie View 5.5). We performed data analysis using a custom Python script. Briefly, we first corrected for photobleaching by fitting the brightness data points with the sum of two exponentials. The fractional fluorescence change (-ΔF/F) corresponds to $-(F(t) - F_0)/F_0$, where F(t) corresponds to the

fluorescence at time t. $F_0$ corresponds to the baseline (resting) fluorescence. All data are corrected for photobleaching. To obtain $F_0$, we averaged 100 data points before application of the stimuli. We filtered out noise by eliminating spikes that were <5% of the signal. Activation latency is the time before the first activation occurred following application of the stimulus.

## Glucose measurements in adult hemolymph

Hemolymph glucose level was measured as described previously (*Dus et al., 2011*). Briefly, to measure circulating glucose, hemolymph samples were collected from ten 3- to 5-day-old males. The animals were punctured in the thorax using a fine injection needle and placed shoulder down to prevent leakage from the genital tract into 0.5 mL tubes whose bases had been punctured with a 21-gauge needle. The tubes were set into 1.5 mL microfuge tubes and centrifuged at 4°C for 5 min at ~2800 $g$ force; 0.5 µL of hemolymph was added to 14.5 µL PBS and placed at 70°C for 5 min. Then, we added 100 µL of glucose reagent (Sigma: G3293 VER) and incubated at 37°C for 12 hr. We used a commercial glucose (HK) assay kit (Sigma: GAHK-20-1KT) to perform the assays and measured the total glucose at 340 nm. Glucose levels were quantified by comparing the values with a glucose standard curve.

## Trehalose and glucose measurements in whole adult flies

Quantification of trehalose and glucose levels in whole fly extracts were performed as described previously (*Meunier et al., 2007*). Briefly, 10 males were weighed and homogenized in 250 µL of 0.25 M $Na_2CO_3$ buffer and incubated in a water bath at 95°C for 5 min to inactivate all enzymes. Then 150 µL 1 M acetic acid and 600 µL 0.25 M sodium acetate (pH 5.2) were added, and the solution was centrifuged (10 min, 12,500 $g$ force, 24°C); 200 µL of each supernatant was incubated overnight at 37°C with 2 µL porcine kidney trehalose (Sigma: T8778 UN) to convert trehalose into glucose, and 100 µL of this solution was added to 1 mL glucose hexokinase solution (Sigma: GAHK-20) and incubated for 20 min at 37°C. Glucose levels were quantified at 340 nm. We determined glucose concentrations using a glucose standard curve.

## Glycogen measurements

We quantified glycogen levels in whole fly extracts as described (*Dus et al., 2011*). Briefly, five males (3–5 days of age) were weighed and homogenized in 100 µL of ice cold phosphate buffered saline (1× PBS). We inactivated the enzymes by incubating the homogenates at 70 °C for 5 min, centrifuged the samples at 12,500 $g$ force for 3 min at 4 °C, and transferred 20 µL supernatants to 1.5 mL tubes, and added 20 µL diluted to 1:3 in 1× PBS; 1.5 µL amyloglucosidase (Sigma A1602) suspension was inserted in 998.5 µL 1× PBS. We added 20 µL of the diluted amyloglucosidase solution to 20 µL aliquots of each test sample (and to glycogen standards) to convert the glycogen into glucose. The samples and glycogen standards were incubated at 37°C for 60 min. A commercial glucose (HK) assay reagent (Sigma: G3293 VER) was used to measure total glucose at 340 nm. To determine the glycogen concentrations, we compared the glucose levels in the test samples with a standard curve derived from converting glycogen standards to glucose.

## Tip recordings

We performed tip recordings essentially as previously described with minor modifications (*Hodgson et al., 1955*; *Moon et al., 2006*). Three- to 5-day-old male or female flies were anesthetized on ice, and a reference electrode containing Ringer's solution was inserted from the thorax up to the tip of labellum. We stimulated L6 sensilla with 50 mM D-glucose or 200 mM L-glucose introduced in a glass recording electrode (tip size 10–20 µm) with 30 mM tricholine citrate as the electrolyte. The recording electrode was connected to a preamplifier (TastePROBE, Syntech, Hilversum, The Netherlands; http://www.ockenfels-syntech.com/), and the signals were collected and amplified 10× using a signal connection interface box (Syntech) in conjunction with a 100–3000 Hz band-pass filter. Recordings of action potentials were made using a 12 kHz sampling rate and analyzed using Autospike 3.1 software (Syntech).

## PER assays

We performed PER assay as described previously (*Shiraiwa and Carlson, 2007*) with slight modifications. Briefly, we cut off the ends of P200 pipette tips to slightly increase the openings. We inserted a

fly that was starved for 5 or 18 hr so that only the head was exposed to the outside. Each fly was then kept in a humidified chamber for 30 min. We offered water to the flies so that they become water saturated. To confirm that they were not responding to water, we re-applied water stimuli (three times in a 2 s interval) to the labellum, and only used animals that were unresponsive to water alone. We then touched the labellum with a Kimwipe soaked in either 50 mM D-glucose or 200 mM L-glucose. The stimuli were applied to the flies a maximum of three times to decide whether they were responsive or non-responsive. If there was a single full extension of the proboscis to the stimuli, this was recorded as a positive PER. The responses were recorded using a Nikon SMZ745 stereomicroscope, and *i*solution IMTcam3 camera and Bandicam software. To quantify the times to full extension of the proboscis, we recorded videos at 8 frames/s (125 ms resolution), and measured the times to fully extend the proboscis after the stimulus was applied to the labellum. We cropped the recorded videos using a daum pot encoder (https://www.daum.net/).

## Ingestion assays

Ingestion assays were performed as previously described with slight modifications (*Sang et al., 2019*). Briefly, five males 3–5-days of age were transferred to food vials with freshly prepared 1% agarose containing 5% sucrose, and blue dye (Brilliant blue FCF, 0.125 mg/mL; Wako Chemical Co., cat. # 027–12842). We allowed the flies to feed on the food for the indicated durations starting at 9 AM (ZT0: lights on). The flies were homogenized in 1.5 mL microfuge tubes in 1 mL distilled water. The tubes were centrifuged for 3 min at 12,500 $g$ force, the supernatants were transferred to cuvettes, and ODs were measured at 629 nm.

## Defecation assays

Defecation assays were performed as described previously (*Dus et al., 2015*). Briefly, 20 male flies (3–5 days of age) were fed food that consisted of 1% agarose mixed with 5% sucrose and blue dye (Brilliant blue FCF, 0.125 mg/mL; Wako Chemical Co., cat. # 027-12842) for 24 hr. The flies were transferred to an empty clean vial for 1 hr. The vials were cleared, and then we determined the number and size of excreta by photographing the bottoms of the vials with an *i*solution IMTcam3 camera connected to a Nikon SMZ745 stereomicroscope. The defecation rate was the number of spots per fly per hour. To measure the diameters of the fecal spots, we magnified the images 400-fold and determined the average area/spot using the area formula for circles. To quantify the total excreta, we washed the inside of each vial with 1 mL 1× PBS and measured the OD of the solution at 629 nm.

## Scoring crop sizes

To score crop sizes, we performed experiments as described previously (*Edgecomb et al., 1994*). Briefly, we placed ~20 males on food that consisted of 1% agarose mixed with 5% sucrose, and blue dye (Brilliant blue FCF, 0.125 mg/mL; Wako Chemical Co., cat. # 027-12842) for 12 hr; 10–20 flies per each group were dissected in 1× PBS under the microscope and the dye-filled crops were scored. We adopted the 1–5 scoring system (*Edgecomb et al., 1994*): (1) the crop shrank and no dye was visible in the crop, (2) a small blue spot was visible in the crop, (3) the crop was wider and it was long but extended toward the lateral side of the abdomen, (4) the crop was round and the abdomen was slightly swollen, and (5) the crop was maximally distended and the abdomen was extremely swollen. The percentages of flies with each crop score were calculated.

## Crop contraction measurements

Crop contraction measurements were conducted as described previously with modifications (*Solari et al., 2017*). Briefly, males were fed 1% agarose mixed with 5% sucrose and food dye overnight in a humidified chamber. Flies were cold anesthetized and fixed on a glass slide with glue. The fixed flies were transferred and submerged in a Petri dish cover with 1× PBS (128 mM NaCl, 36 mM sucrose, 4 mM $MgCl_2$, 2 mM KCl, 1.8 mM $CaCl_2$, pH 7.1). The ventral cuticle was removed gently to view the crop, and the number of crop contraction per 30 s was counted using a Bandicam Screen to record videos, we use an *i*solution IMTcam3 camera connected to a Nikon SMZ745 stereomicroscope.

## Weight measurements

To measure the weights of flies before and after starvation, we used animals 3–4 days post-eclosion. The weight of groups of 10 flies were measured under sated conditions, and then again after 24 hr of starvation. To determine the weights, we anesthetized sated flies, transferred them to a 1.5 mL tube, and quickly measured their weights using a digital balance (OHAUS Corporation; AVG264), which provides $10^{-4}$ g sensitivity. We subtracted the weights of the tubes only from the total weight. The same groups of 10 flies were starved in vials containing damp Kimwipe paper. After 24 hr, the body weights of the starved flies were determined.

## Statistical analyses

No statistical methods were used to pre-determine sample sizes. We used similar sample sizes as has been reported in previous publications from multiple research groups considering the variance and effect size in data variability. The experiments were conducted on multiple days, and the data were analyzed using GraphPad Prism software. The sample size for each experiment is reported in the corresponding figure legend. In most cases, raw values are displayed on the graph plots, which provide the sample sizes. The statistical test performed on each dataset is described in the corresponding figure legend. All data are presented as means ± SEMs. Single-factor ANOVA with Scheffe's analysis was used as a post hoc test to compare multiple sets of data for most experiments except in *Figure 2*. The $Ca^{2+}$ responses shown in *Figure 2* were analyzed using the Mann-Whitney test. The data shown in *Figure 6F* were analyzed using unpaired Student's t-tests. Black and red asterisks indicate statistical significance from the controls and mutants, respectively. \*\*$p < 0.01$. All statistical values are displayed in the source data file.

## Group allocation

All the experimental groups of flies were reared under the same conditions. The flies were collected on the same day, were the same ages, and the experiments were performed on the same day.

## Acknowledgements

We thank Dr Dhananjay Thakur for valuable intellectual contributions. This work was supported by grants to YL from the Basic Science Research Program of the National Research Foundation of Korea (NRF) funded by the Ministry of Education (NRF-2016R1D1A1B03931273, NRF-2018R1A2B6004202, and NRF-2021R1A2C1007628) and the Korea Environmental Industry and Technology Institute (KEITI) grant funded by the Ministry of Environment of Korea, and by grants to CM from the NIDCD (DC007864) and the NIAID (AI65575 and AI169386). SD was supported by the Global Scholarship Program for Foreign Graduate Students at Kookmin University in Korea.

## Additional information

### Funding

| Funder | Grant reference number | Author |
| --- | --- | --- |
| National Institute on Deafness and Other Communication Disorders | DC007864 | Craig Montell |
| National Institute of Allergy and Infectious Diseases | AI165575 | Craig Montell |
| National Institute of Allergy and Infectious Diseases | AI169386 | Craig Montell |
| National Research Foundation of Korea | NRF-2018R1A2B6004202 | Youngseok Lee |
| National Research Foundation of Korea | NRF-2016R1D1A1B03931273 | Youngseok Lee |

| Funder | Grant reference number | Author |
|---|---|---|
| National Research Foundation of Korea | NRF-2021R1A2C1007628 | Youngseok Lee |
| Korea Environmental Industry and Technology Institute | | Youngseok Lee |

The funders had no role in study design, data collection and interpretation, or the decision to submit the work for publication.

### Author contributions

Subash Dhakal, Qiuting Ren, Jiangqu Liu, Data curation, Formal analysis, Investigation, Methodology, Writing – review and editing; Bradley Akitake, Data curation, Formal analysis, Investigation, Methodology; Izel Tekin, Investigation, Methodology; Craig Montell, Youngseok Lee, Conceptualization, Formal analysis, Funding acquisition, Project administration, Supervision, Writing – original draft, Writing – review and editing

### Author ORCIDs

Craig Montell (ID) http://orcid.org/0000-0001-5637-1482
Youngseok Lee (ID) http://orcid.org/0000-0003-0459-1138

### Decision letter and Author response

Decision letter https://doi.org/10.7554/eLife.56726.sa1
Author response https://doi.org/10.7554/eLife.56726.sa2

## Additional files

### Supplementary files

• Transparent reporting form
• Source data 1. Detailed statistics for all data.

### Data availability

All data generated or analysed during this study are included in the manuscript and supporting files. Source data files have been provided for Figures 1-7, and Figure supplements 1-7.

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
