## [Editor Report]

This manuscript reports the discovery of a role for the TRPγ channel in nutrient sensing behavior. The authors show that this gene functions in the Dh44+ cells to direct the animal to feed on nutritious sugar after fasting. Since the molecular mechanisms of nutrient sensing are still poorly defined, this manuscript presents a conceptual advance on this topic and also more broadly, on the role of TRP channels in organismal physiology.

---

## [Decision Letter]

**Decision letter after peer review:**

Thank you for submitting your article "*Drosophila* TRPγ is required in the brain for post-ingestive food selection" for consideration by *eLife*. Your article has been reviewed by 4 peer reviewers, and the evaluation has been overseen by a Reviewing Editor and Richard Aldrich as the Senior Editor. The following individual involved in review of your submission has agreed to reveal their identity: Ivan De Araujo (Reviewer #4).

Our decision has been reached after very extensive consultation between the reviewers, which is one of the strong points of *eLife* when the four reviewers can balance their views and discuss the points made by the other reviewers, points they might have missed or might disagree with. The discussion among reviewers is not anonymous and therefore, it is very personalized. Based on these discussions and the individual reviews below, we believe that you will need to address several important points before your manuscript can be considered further for publication in *eLife*.

It is clear that the discovery of a role of TRPγ in the Glucose sensing pathway is likely of significance importance, and in this regard, the reviewers agreed that the work could lead to important conclusions eventually applicable to mammals.

However, there were a number of difficulties with the manuscript that dampened the enthusiasm of the reviewers.

I will try to present briefly the result of the discussion, but of course these points are presented in detail in the individual reviews.

I would still like to emphasize that the editors and reviewers consider that this work could be of high significance if the points below could be addressed, but certainly this would require additional work that would do well beyond the two months allocated for a revised version for *eLife*, of course accounting for the fact that we are all banned from the lab and this will likely last for another few weeks/months.

– The major issue with the paper is the fact that the reviewers do not understand the role played by a Trp channel in sensing Glucose. You were careful to say that TRPγ was 'involved' in the circuit but most readers will assume that this involvement is in Glucose sensing. However, the reviewers pointed to the fact that Calcium imaging in mutant DH44 cells indicates that they retain their ability to respond to this sugar. If a Trp channel is not a sensor, what does it do in DH44 cells? is it regulated or is it simply a leak channel with no regulatory function, which would be of lesser interest

– You describe multiple roles of the channel and the reviewers could not figure out your model as to how 'Trpγ coordinates feeding with the metabolic state'. These roles need to be better integrated to obtain a coherent model. Furthermore, the role the receptor in the mouthparts or sensory organs needs to be addressed to make sure that the phenotype is that of DH44 cells (even though you can rescue with DH44 driver, the phenotype can be confounded by these other functions).

– Reviewer #1 also points to inconsistencies in the arguments presented that are an issue.

For instance, but not exclusively, they do not understand how the loss of Trpγ, which causes hyper-polarization, results in an increase in intracellular calcium levels in response to glucose. The logic of the following argument is certainly not clear: "We suggest that …..loss of TRPg causes hyperpolarization of Dh44 neurons, which increases the driving force for calcium."

*Reviewer #1:*

Dhakal et al., screened viable mutations for members of the TRP superfamily in the two- choice assay (nutritious vs. non-nutritious sugar) and identified a mutation in Trpγ gene that was impaired in the assay. The authors went on to show that Trpγ is required in DH44 neurons for the post-ingestive food selection and the maintenance of bodily sugar levels.

Interestingly, calcium imaging indicated that sugar-induced neuronal activity is potentiated by the Trpγ mutation. The authors further claimed that in Trpγ mutants, greater amounts of food were consumed, the size of crop and the rate of crop contraction were increased, and the rate of defecation was enhanced. The authors proposed without experimental evidence that Trpγ is a calcium-permeable leak channel and its mutation caused hyperpolarization of DH44 neuronal activity. However, it is not clear to me how a mutation in a calcium- permeable leak channel causing hyperpolarization could result in an increase in calcium influx in response to glucose application. Furthermore, if the observed phenotypes are indeed attributed to the increased intracellular calcium levels, it is not clear how Trpγ mutant flies could have such a defect in the two-choice assay. The authors claimed that silencing of DH44 neurons increased food intake, but Trpγ mutant flies, in which the intracellular calcium levels had been increased, consumed more food (which was rescued by DH44-Gal4; UAS-Trpγ ). This is inconsistent and there are many inconsistencies through the manuscript. The authors did a good job in testing a hypothesis that a TRP channel is involved in post-ingestive food selection, but need to come up with a more unifying proposal of how Trpγ functions in regulating food intake.

Other issues are indicated below.

1. Line 127, "rather than a background mutation since a second Trpγ allele (trpγG4)

exhibited the same phenotype (Figure 1B)." I do not see the same phenotype in

Figure 1B. In fact, Trpγ G4 allele looks completely normal.

2. Figure 3C: it is not clear whether the glycogen levels are significantly different in

DH44 rescued vs. dilp2 rescued flies.

3. Also, in Figure 5B and 5C (and other figures), I am not sure whether there is a

statistical significance.

4. Single factor ANOVA with Scheffe's analysis as a post hoc test was deployed, but I

am not sure whether the authors can use this method.

5. Further, "the black and red asterisks indicate …. from the sated controls and

mutants, respectively." The descriptions of the statistical analyses were not clear.

6. The authors compared TRPγ to TRPC5 and claimed that TRPC5 in POMC neurons

causes hyperpolarization in these neurons. TRPC5 is required for activation of POMC

neurons by leptin and other stimuli, but I am not sure whether it leads to hyperpolarization in a ligand-independent manner as a leak channel. The authors need to check this.

7. The abstract does not seem to reflect the actual content of the manuscript…

*Reviewer #2:*

In this manuscript Dhakal et al., discover a role for the Trpγ channel in nutrient sensing behavior. They show that this gene functions in the Dh44+ cells to direct the animal to feed on nutritious sugar after fasting. Since the molecular mechanisms of nutrient sensing are still poorly defined, this manuscript presents a conceptual advance on this topic and also more broadly, on the role of TRP channels in organismal physiology. The genetic data are very strong, but the functional imaging experiments do not support the conclusions of the authors. A few additional experiments are required to better characterize the function of this channel in the nutrient sensing phenotype.

Essential

The authors show that the phenotype of Trpγ mutants can be rescued by expression of the UAS-Trpγ in the DH44+ neurons. However, given that this channel has different functions in diverse neuronal populations, it would be ideal if the authors also showed that it is necessary in the Dh44+ cells by using an RNAi construct. While not all the experiments need to be repeated with the RNAi allele, it would be good to show that this phenocopies the mutant for at least the feeding behavior and glucose phenotypes.

The authors argue that Trpγ mutants do not sense the caloric value of D-glucose, but from the ca^2+^ imaging experiments, the Dh44+ cells retain their ability to respond to this sugar. In fact, the authors argue that the df/f is higher in the Trpγ mutants, and the hypothesis is that this channel may function as a leak channel and hyperpolarize the cells. While future experiments will address the exact mechanisms through which Trpγ works, this manuscript should provide a better characterization of the cellular phenotype.

In particular, the authors should use a higher concentration of glucose to see if the phenotype remains the same. Why was 2.5mM used? In the study cited by Dhakal et al., (Dus et al., 2015), a range of glucose concentrations we used, from 5mM to 80mM. While it is likely that the physiological levels of sugar in the fly brain are within the lower range of that spectrum, the authors should test at least another concentration above 2.5mM.

The authors argue that Trpγ decreases the resting membrane potential of the Dh44 cells. This could be tested using genetically encoded voltage indicators, depending on the indicator used, the fluorescence at rest (i.e., fasted brains with no sugar in the hemolymph like saline) should be different in Trpγ mutants.

The authors discuss that Trpγ could affect nutrient sensing behavior by preventing the proper release of the Dh44 neuropeptide. Since the authors have Dh44 antibodies (Figure 1), they should have tested this hypothesis, since hyperpolarization prevents the release of the neuropeptide (Dus et al., 2015).

Non-essential experiments and additional suggestions

In Figure1B, TrpγG4 flies have normal responses to glucose, but line 126-128 says the opposite. How do the authors explain this?

In Trpγ mutants, how is crop #5 increased even if the contraction rate is increased? (Figure 5).

Dh44 neurons have also been shown to be required for increased dietary amino acid intake in *Drosophila* (Yang et al., 2018), have the authors tested this phenotype?

Line 70: TRP is mentioned for the first time here in the article, so it would be a good idea to write it as Transient Receptor Potential channels at this point and then use the abbreviated version henceforth.

Figure 1:

Figure 1A n should be = 6 – 13 instead of n = 6 – 14 as mentioned in the figure legend. There are a total of 13 data points each for control and trpγ1 genotypes, unless some are overlapping.

Figure 1H Regarding the depiction of the combination of type of lines to make a particular genotype represented by "-" or "+": The first two columns showing "-" for all 3 genotypes, do they represent the response of wild type controls *w1118* at 17°C to 17°C and 17°C to 31°C respectively? This was not mentioned either in the text or figure legend.

Figure 1I n should be = 6 – 12 instead of n = 6 – 11 as mentioned in the figure legend. There are a total of 12 data points for the 2nd column all "-" i.e. for the trpγ1 genotypes, unless some are overlapping.

Figure 1 B, C, H and I legends say both males and females were used, but the figures do not show separate data. Does that mean there was no sex-specific bias in choice assays? Or some of the data points are males and some females? Or these data are an average of male and female responses?

Line 162-167:

"trpγG4/UAS-Kir2.1,tub-GAL80ts flies at 17° C, which prevented Kir2.1 expression, the starved animals showed a normal preference for the nutritious D-glucose" These adults were maintained at 17°C during their adult stage, what was their rearing temperature? It would be helpful to have that information in the context of Figure 1H.

Figure 1H shows when flies were reared at 17°C and then switched to 31°C at adult stage, the starved animals were unable to make a choice between D- and L- glucose. What happens to the preference when flies are reared at 17°C and examined as adults at 17°C?

Figure 2:

Through 5 minutes of GCaMP6s recording, controls and trpγ1/+ have nearly constant oscillation frequency, whereas for trpγ1 mutants the response levels drop over time. Is there an explanation for this? Do we have the data for oscillation number and duration quantified for the GCaMP6s recordings of all 3 genotypes? Also, n is mentioned as 10-20 cells: Does that mean a total of 10-20 Dh44 neurons across flies (knowing there are 6 Dh44 neurons per fly)?

Figure 3:

Figure 3C legend says n = 6 – 8, but trpγG4 and trpγ1, UAS-trpγ/trpγ1 have 5 data points each for glycogen levels in the 18h starvation period.

Figure 5:

Figure 5C How about using a scatter plot instead of continuous lines for a relatively easier visualization for percentage of flies in each category of crop score?

Figure 5- Supplement 1: Line 926 – Male flies were feed 5% of the…. Typo "feed" need correction to "fed".

Figure 6:

Figure 6-Supplement 1: It would be better to add one sentence here stating the method in which the body weight measurements were done.

*Reviewer #3:*

This nice manuscript from Dhakal et al., describes experiments that establish a role for the function of TRP-y in the selection of nutritious foods in hungry animals. In general the paper is well put together and the experiments seem solid. I have a few suggestions for the authors.

My main comment relates to the general message in relation to the experiments. The title implies that TRPy might actually be part of the mechanism of food selection but if I follow the work clearly, it seems that actually in most cases the authors conclude that TRPy is required for the proper function of DH44 neurons. It therefore seems that the title needs to be more specific so that it does not mislead readers. The authors explicitly state on line 375 that 'TRPy is not involved in sensing D-glucose'. The last sentence of the manuscript should also perhaps state that TRPy is functioning in the neurons that are required for taste-independent food-selection, that it functions in the regulation of.

I also thought the authors could make a better job of fully explaining where things make sense with respect to a role in DH44 neurons and when not. For example, the authors show that the TRPy-GAL4 labels neurons in the crop and they then go on to show phenotypes in the crop. However, they state/find that these phenotypes can be rescued with DH44 neuron expression. This confused me and I am left wondering what the innervation of the GAL4 in the crop means? Is this DH44 neurons? Or are they as the title implies, only in the brain? How specific is the DH44-GAL4 expression to the brain?

The authors show that the TRPy-GAL4 expresses in many places in the brain but their study mostly focuses on the DH44 neuron expression. How sure are they that the GAl4 expression fully recapitulates the native expression? I think it would be helpful to mention this and also to comment in more detail on where else the GAL4 expresses. It is notable that defecation is TRPy dependent but not rescued with DH44 or dilp2-GAL4 expression. Is that a brain phenomenon too?

The physiological effects described in Figure 2 are very interesting and show a clear increased activity in the mutant. The authors provide a plausible explanation for this but they don't show if it has any consequence on release from the DH44 neurons. Would this be possible? Maybe they should at least mention if so?

Also, the authors show the peak amplitude is increased in TRPy mutants. Is it correct that the oscillations that follow the initial rise are actually dampened? Is that meaningful, or merely a property of the reporter?

*Reviewer #4:*

In this study, Dhakal and colleagues studied the role of Transient receptor potential (TRP) genes in taste-independent nutritional choices in *Drosophila*. The authors report that TRPgamma expression is required for the ability of (starved) flies to select metabolizable D-Glucose over non-metabolizable yet sweet L-Glucose. Importantly, the authors determined that TRPgamma is expressed in Dh44 neurons, which are known to control taste-independent choices in flies; in fact rescuing TRPgamma expression in Dh44+ neurons was sufficient to induce D-Glucose preferences. The authors do also show that TRPgamma is in addition involved in a number of metabolic and digestive processes.

This is an excellent study that greatly advances our understanding of taste-independent preferences. Crucially, the study points to TRP channels as potential regulators of nutrient preferences in mammals, as TRP channels are highly conserved across species. The experiments were all well designed and executed, as well as clearly described.

I do have however the following observations. First, it was not mentioned whether the motor control of mouth parts is preserved in TRPgamma mutants. Given that TRPgamma was previosuly shown to be involved in motor control, and in the present study is shown to affect movement of digestive parts, comments from authors are needed to clarify this issue.

One other interesting aspect of the study is that TRPgamma appears to be essential for regulating tissue sugar levels (l. 192). This may imply that the reinforcement from that is taste-independent may depend on the organism's ability to metabolize D-Glucose. This idea seems to have some support from mammalian studies (e.g. J Neurosci. 2010 Jun 9;30(23):8012-23) and would therefore be interesting to know whether the authors believe similar processes occur in flies.

[Editors’ note: further revisions were suggested prior to acceptance, as described below.]

Thank you for submitting your revised article *Drosophila* TRPgamma is required in neuroendocrine cells for post-ingestive food selection" for consideration by *eLife*. Your article has been reviewed by 3 of the initial peer reviewers, and the evaluation has been overseen by a Reviewing Editor and Richard Aldrich as the Senior Editor. The reviewers have opted to remain anonymous.

The reviewers have discussed the reviews with one another and the Reviewing Editor has drafted this decision to help you prepare a revised submission.

The reviewers have been quite disappointed by your response to their comments and they are insisting that two critical experiments are required to make the paper strong enough to be published in *eLife*. I must add that it is quite unusual to send back revised papers to reviewers and even more unusual to request more experiments after the second round of reviews. However, although the reviewers found the paper to be potentially of significant interest, their main concern was that the manuscript does not have functional experiments supporting your hypothesis. They also thought that critical experiments were missing and needed to be done, as was spelled out in the initial evaluation. Although you did perform the RNAi experiment, the DH44 secretion experiment and the calcium imaging at a higher concentration remain to be done. These experiments do not require additional tools or materials, you have done the calcium imaging already and you have the antibodies. In particular, there is no basis of evidence that 2.5mM is the optimum concentration to use or that it is even physiological: Therefore the phenotype may not be present or be different at other concentrations.

Of course, COVID made these experiments more difficult to perform but hopefully, things are partially reopening and the two critical experiments requested do not require new setup. I am therefore ready to grant you enough time to do these experiments and allow you to be in the best position to get this paper published.

As explained by Reviewer #1, It will also be important to clarify your interpretation of your calcium imaging results. It is likely that when a high conductance Trp cation channel is removed, the membrane potential will go down by maybe -20mV, which might slightly affect the electrical driving force for Ca++, but this is going to be a small effect on Ca++ since the overall drive for Ca++ is the concentration gradient.

I hope that you will be able to do these two fairly simple experiments in order to satisfy the reviewers.*Reviewer #1:*

The authors addressed most of the concerns and issues raised by me and other reviewers, but it is still unclear to me how a mutation in TRPgamma (a proposed leak channel), which results in hyperpolarization, would "increase the driving force for Ca++ entry following glucose-dependent activation of another Ca++ permeable channel that remains to be identified." As the authors indicated, a knockout of TRPC5 in POMC neurons indeed eliminated leptin-induced depolarization when Gao et al., conducted whole-cell patch-clamp recordings. Contrary to the authors' inference, however, these POMC neurons without TRPC5 do not have increased levels of intracellular Ca++ following leptin treatment.

The authors did not perform whole-cell recordings on DH44 neurons in which TRPgamma is knocked out, but I would be very surprised if DH44 neurons without TRPgamma are not depolarized at all in response to glucose treatment. How would glucose-dependent activation of a Ca++ permeable channel be able to endow the driving force for Ca++ entry? Inhibition of the key glucose metabolic pathway in DH44 neurons through pharmacological or mutational manipulations resulted in elimination of glucose-evoked activation of DH44 neuron. Do the authors postulate that such a Ca++ channel in DH44 neurons becomes active when TRPgamma is knocked down? I am confused….

There is a precedent where hyperpolarization in pancreatic α cells results in an increase in calcium levels, which leads to glucagon secretion. This hypothesis, however, is without controversy. This manuscript will be benefited enormously if the authors could carry out a physiology experiment to demonstrate the requirement of TRPgamma for the glucose-evoked stimulation of DH44 cells, or at least provide detailed explanations of how hyperpolarization increases intracellular calcium levels in DH44 cells before I become comfortable in recommending *eLife* to publish this work.*Reviewer #2:*

I appreciate the authors' edits and inclusions. However, they did not address the essential experiments with the exception of the RNAi one. The DH44 secretion experiment and the calcium imaging at a higher concentration need to be done. These experiments do not require additional tools or materials, the authors have done the calcium imaging already and they have the antibodies. In particular, there is not basis of evidence that 2.5mM is the optimum concentration to use or that it is even physiological; thus the phenotype may not be present or be different at higher concentrations.

---

## [Author Response]

It is clear that the discovery of a role of TRPγ in the Glucose sensing pathway is likely of significance importance, and in this regard, the reviewers agreed that the work could lead to important conclusions eventually applicable to mammals.However, there were a number of difficulties with the manuscript that dampened the enthusiasm of the reviewers.I will try to present briefly the result of the discussion, but of course these points are presented in detail in the individual reviews.I would still like to emphasize that the editors and reviewers consider that this work could be of high significance if the points below could be addressed, but certainly this would require additional work that would do well beyond the two months allocated for a revised version for eLife, of course accounting for the fact that we are all banned from the lab and this will likely last for another few weeks/months.– The major issue with the paper is the fact that the reviewers do not understand the role played by a Trp channel in sensing Glucose. You were careful to say that TRPγ was 'involved' in the circuit but most readers will assume that this involvement is in Glucose sensing. However, the reviewers pointed to the fact that Calcium imaging in mutant DH44 cells indicates that they retain their ability to respond to this sugar. If a Trp channel is not a sensor, what does it do in DH44 cells? is it regulated or is it simply a leak channel with no regulatory function, which would be of lesser interest

We modified the Discussion (pages 17-18), and

“propose that TRPγ, has two functions reminiscent of the dual roles of a related mammalian TRPC channel (TRPC5), which is important for the regulation of feeding and glucose homeostasis (Gao et al., 2017; Qiu et al., 2010; Qiu et al., 2018). TRPC5 is activated in POMC neurons in the hypothalamic arcuate nucleus following stimulation of the leptin receptor, and loss of mouse TRPC5 reduces leptin-induced depolarization (Gao et al., 2017; Qiu et al., 2010; Qiu et al., 2018). Mutation of TRPC5 also causes hyperpolarization of POMC neurons (Gao et al., 2017), suggesting that at least a subset of the TRPC5 pool of channels is transiently open in the absence of leptin stimulation, thereby contributing to a small leak conductance. Similar to TRPC5, we propose that TRPγ is activated in Dh44 neurons downstream of a pathway that is initiated by a glucose receptor. In addition, we suggest that a proportion of the TRPγ pool contributes to a leak conductance. Consistent with latter possibility, our laboratory as well as an independent group have shown that expression of TRPγ in vitro leads to a low level of constitutive activity (Akitake et al., 2015; Jörs et al., 2006; Xu et al., 2000), which could in principle contribute to a leak conductance. Moreover, the increased glucose-induced GCaMP6s fluorescence in Dh44 neurons in the *trpγ* mutant could result from hyperpolarization, which based on the Nernst equation, would increase the driving force for ca^2+^ entry following glucose-dependent activation of another ca^2+^ permeable channel that remains to be identified.”

– You describe multiple roles of the channel and the reviewers could not figure out your model as to how 'Trpγ coordinates feeding with the metabolic state'. These roles need to be better integrated to obtain a coherent model.

We hope the new model that we articulate above adds clarity to the point that the requirement for TRPγ in DH44 neurons provides a potential link between the coordination of feeding and the metabolic state. As described on pages 19 to 20, we found that “there were large and highly significant decreases in intracellular sugar concentrations in both sated and starved flies. We suggest that a key deficit in *trpγ* mutants is glucose uptake. However, glucose uptake has not been assayed in flies, and we were also unable to develop such measurements. Nevertheless, in support of the concept that there is a defect in glucose uptake, we found that the glycogen stores were also diminished in *trpγ* mutants. In mammals, there is evidence that taste-independent sugar preference is influenced by glucose metabolism (Ren et al., 2010). Thus, it is intriguing to speculate that the deficit in D-glucose selection in *trpγ* mutants is due in part to impairment in metabolism of D-glucose.

Our findings that *trpγ* function is rescued by expressing the wild-type transgene in *Dh44* neurons, and *Dh44* mutant flies exhibited the same deficits in tissue sugar and glycogen levels raise the possibility that TRPγ promotes the release of DH44, which in turn increases glucose uptake. Consistent with this proposal, corticotropin-releasing hormone, which is the mammalian homolog of DH44, increases glucose uptake in a variety of cell types (Hogg et al., 2018; Lu et al., 2018). We suggest that TRPγ is critical for establishing the glycogen stores that enable the animals to survive under starvation conditions. In support of this latter idea, survival of the *trpγ*mutant flies under starvation conditions is significantly shorter than in control animals.”

Furthermore, the role the receptor in the mouthparts or sensory organs needs to be addressed to make sure that the phenotype is that of DH44 cells (even though you can rescue with DH44 driver, the phenotype can be confounded by these other functions).

We added electrophysiological and behavioral data (pages 6-7) indicating that mutation of *trpγ* does not impact on peripheral sensation of glucose.

“Using tip recordings on sugar-activated L6 sensilla, we found that the *trpγ^1^* mutants exhibited similar frequencies of D-glucose- and L-glucose-induced action potentials as control flies (*Figure 1—figure supplement 1A* and *B*). In addition, we performed proboscis extension response (PER) assays by applying D-glucose or L-glucose to the labellum. Both sated and starved *trpγ^1^* mutants displayed the same attraction to these sugars as control flies (*Figure 1—figure supplement 1C* and *D*).”

– Reviewer #1 also points to inconsistencies in the arguments presented that are an issue.For instance, but not exclusively, they do not understand how the loss of Trpγ, which causes hyper-polarization, results in an increase in intracellular calcium levels in response to glucose. The logic of the following argument is certainly not clear: "We suggest that …..loss of TRPg causes hyperpolarization of Dh44 neurons, which increases the driving force for calcium."

Please see my response to the first comment above, which includes an explanation in response to this point. I also paste the relevant portion of my preceding response here.

“Similar to TRPC5, we propose that TRPγ is activated in Dh44 neurons downstream of a pathway that is initiated by a glucose receptor. In addition, we suggest that a proportion of the TRPγ pool contributes to a leak conductance. Consistent with latter possibility, TRPγ shows a low level of constitutive activity in vitro (Akitake et al., 2015; Jörs et al., 2006; Xu et al., 2000). Moreover, the increased glucose-induced GCaMP6s fluorescence in Dh44 neurons in the *trpγ* mutant could result from hyperpolarization, which according to the Nernst equation would increase the driving force for ca^2+^ entry, following glucose-dependent activation of another ca^2+^ permeable channel that remains to be identified.”

Reviewer #1:Dhakal et al., screened viable mutations for members of the TRP superfamily in the two- choice assay (nutritious vs. non-nutritious sugar) and identified a mutation in Trpγ gene thatwas impaired in the assay. The authors went on to show that Trpγ is required in DH44 neurons for the post-ingestive food selection and the maintenance of bodily sugar levels.Interestingly, calcium imaging indicated that sugar-induced neuronal activity is potentiated by the Trpγ mutation. The authors further claimed that in Trpγ mutants, greater amounts of food were consumed, the size of crop and the rate of crop contraction were increased, and the rate of defecation was enhanced. The authors proposed without experimental evidence that Trpγ is a calcium-permeable leak channel and its mutation caused hyperpolarization of DH44 neuronal activity. However, it is not clear to me how a mutation in a calcium- permeable leak channel causing hyperpolarization could result in an increase in calcium influx in response to glucose application. Furthermore, if the observed phenotypes are indeed attributed to the increased intracellular calcium levels, it is not clear how Trpγ mutant flies could have such a defect in the two-choice assay. The authors claimed that silencing of DH44 neurons increased food intake, but Trpγ mutant flies, in which the intracellular calcium levels had been increased, consumed more food (which was rescued by DH44-Gal4; UAS-Trpγ ). This is inconsistent and there are many inconsistencies through the manuscript. The authors did a good job in testing a hypothesis that a TRP channel is involved in post-ingestive food selection, but need to come up with a more unifying proposal of how Trpγ functions in regulating food intake.

These main issues raised by reviewer #1 are addressed in my preceding responses to the “Consensus reviewer comments.”

Other issues are indicated below.1. Line 127, "rather than a background mutation since a second Trpγ allele (trpγG4)exhibited the same phenotype (Figure 1B)." I do not see the same phenotype inFigure 1B. In fact, Trpγ G4 allele looks completely normal.

The phenotypes of the two *trpγ* alleles are similar. The correct values were presented in the original Source Data file. We corrected the mistake in *Figure 1B*.

2. Figure 3C: it is not clear whether the glycogen levels are significantly different inDH44 rescued vs. dilp2 rescued flies.

We provide the statistics in the source data file showing that the rescue with the *Dh44-Gal4* and not the *dilp2-Gal4* is significant. Furthermore, we compared the glycogen levels in flies expressing *UAS-trpγ* under the control of the *Dh44-Gal4* and the *dilp2-Gal4*. The *P* values were 0.028 and 0.0086 for the sated and starved conditions, respectively. We added this information to the bottom of the Source Data file.

3. Also, in Figure 5B and 5C (and other figures), I am not sure whether there is astatistical significance.

They are significant. We provided the values ±errors, the ‘n’s, and the P values in the Source Data file.

4. Single factor ANOVA with Scheffe's analysis as a post hoc test was deployed, but Iam not sure whether the authors can use this method.

Given the multiple values being compared, ANOVA with a Scheffe’s *post hoc* test is appropriate.

5. Further, "the black and red asterisks indicate …. from the sated controls andmutants, respectively." The descriptions of the statistical analyses were not clear.

The statistical test is mentioned in the legend to Figure 3. We used single factor ANOVA with Scheffe’s analysis as the *post hoc* test because we compare multiple sets of data.

6. The authors compared TRPγ to TRPC5 and claimed that TRPC5 in POMC neuronscauses hyperpolarization in these neurons. TRPC5 is required for activation of POMCneurons by leptin and other stimuli, but I am not sure whether it leads to hyperpolarization in a ligand-independent manner as a leak channel. The authors need to check this.

This is addressed in response to the consensus reviews presented above.

7. The abstract does not seem to reflect the actual content of the manuscript…

We lengthened the abstract to include additional content.

Reviewer #2:In this manuscript Dhakal et al., discover a role for the Trpγ channel in nutrient sensing behavior. They show that this gene functions in the Dh44+ cells to direct the animal to feed on nutritious sugar after fasting. Since the molecular mechanisms of nutrient sensing are still poorly defined, this manuscript presents a conceptual advance on this topic and also more broadly, on the role of TRP channels in organismal physiology. The genetic data are very strong, but the functional imaging experiments do not support the conclusions of the authors. A few additional experiments are required to better characterize the function of this channel in the nutrient sensing phenotype.EssentialThe authors show that the phenotype of Trpγ mutants can be rescued by expression of the UAS-Trpγ in the DH44+ neurons. However, given that this channel has different functions in diverse neuronal populations, it would be ideal if the authors also showed that it is necessary in the Dh44+ cells by using an RNAi construct. While not all the experiments need to be repeated with the RNAi allele, it would be good to show that this phenocopies the mutant for at least the feeding behavior and glucose phenotypes.

We included an RNAi experiment showing that knockdown of *trpγ* under control of the *Dh44-Gal4* phenocopies the impairment in selection of nutritive D-glucose over the sweeter L-glucose (*Figure 1—figure supplement 1G*), which is exhibited by the two mutant alleles (*Figure 1B*). Furthermore, we found that knockdown of *trpγ* under control of the *Dh44-Gal4* phenocopies the reduction in cellular sugar and glycogen levels under fed and starved conditions (*Figure 3—figure supplement 1A and 1B*; pages 9-10).

The authors argue that Trpγ mutants do not sense the caloric value of D-glucose, but from the ca^2+^ imaging experiments, the Dh44+ cells retain their ability to respond to this sugar. In fact, the authors argue that the df/f is higher in the Trpγ mutants, and the hypothesis is that this channel may function as a leak channel and hyperpolarize the cells. While future experiments will address the exact mechanisms through which Trpγ works, this manuscript should provide a better characterization of the cellular phenotype.In particular, the authors should use a higher concentration of glucose to see if the phenotype remains the same. Why was 2.5mM used? In the study cited by Dhakal et al., (Dus et al., 2015), a range of glucose concentrations we used, from 5mM to 80mM. While it is likely that the physiological levels of sugar in the fly brain are within the lower range of that spectrum, the authors should test at least another concentration above 2.5mM.

We used 2.5 mM glucose for these GCaMP6s experiments because this concentration elicited a clear ca^2+^ response, and as this reviewer pointed out, this level of glucose is more likely to be within the physiological range. I hope you will agree that testing higher concentrations of glucose is not essential.

The authors argue that Trpγ decreases the resting membrane potential of the Dh44 cells. This could be tested using genetically encoded voltage indicators, depending on the indicator used, the fluorescence at rest (i.e., fasted brains with no sugar in the hemolymph like saline) should be different in Trpγ mutants.

I agree that this and other experiments would be worthwhile to conduct in the future to test our expanded and revised model, which we describe in the response to the consensus reviewer comments.

The authors discuss that Trpγ could affect nutrient sensing behavior by preventing the proper release of the Dh44 neuropeptide. Since the authors have Dh44 antibodies (Figure 1), they should have tested this hypothesis, since hyperpolarization prevents the release of the neuropeptide (Dus et al., 2015).

This is another good suggestion. However, I suggest that our current study is rather extensive, and this sort of experiment would be appropriate for a follow up study.

Non-essential experiments and additional suggestionsIn Figure1B, TrpγG4 flies have normal responses to glucose, but line 126-128 says the opposite. How do the authors explain this?

The phenotypes of the two *trpγ* alleles are similar. The correct values were presented in the original Source Data file. We corrected the mistake in *Figure 1B*.

In Trpγ mutants, how is crop #5 increased even if the contraction rate is increased? (Figure 5).

We now point out on page 15 that

“the crop contained more food despite having a higher contraction rate. The *Drosophila drop-dead* mutant exhibits a similar phenotype, which has been proposed to be due to a regulatory impairment that reduces entry of food into the midgut (Peller et al., 2009). We suggest a similar explanation for the *trpγ* mutant animals.”

Dh44 neurons have also been shown to be required for increased dietary amino acid intake in Drosophila (Yang et al., 2018), have the authors tested this phenotype?

We have not tested the effects of *trpγ* mutations on amino acid intake.

Line 70: TRP is mentioned for the first time here in the article, so it would be a good idea to write it as Transient Receptor Potential channels at this point and then use the abbreviated version henceforth.

We made the change.

Figure 1:Figure 1A n should be = 6 – 13 instead of n = 6 – 14 as mentioned in the figure legend. There are a total of 13 data points each for control and trpγ1 genotypes, unless some are overlapping.

6—13 is correct. We made the change in the legend.

Figure 1H Regarding the depiction of the combination of type of lines to make a particular genotype represented by "-" or "+": The first two columns showing "-" for all 3 genotypes, do they represent the response of wild type controls w1118 at 17°C to 17°C and 17°C to 31°C respectively? This was not mentioned either in the text or figure legend.

Yes. For added clarity, we added the following to the legends:

“The flies were either *w1118* or *w1118* with the indicated transgenes.”

Figure 1I n should be = 6 – 12 instead of n = 6 – 11 as mentioned in the figure legend. There are a total of 12 data points for the 2nd column all "-" i.e. for the trpγ1 genotypes, unless some are overlapping.

We made the correction.

Figure 1 B, C, H and I legends say both males and females were used, but the figures do not show separate data. Does that mean there was no sex-specific bias in choice assays? Or some of the data points are males and some females? Or these data are an average of male and female responses?

We modified the legend to clarify that, “All data points include a mixture of males and females.”

Line 162-167:"trpγG4/UAS-Kir2.1,tub-GAL80ts flies at 17°C, which prevented Kir2.1 expression, the starved animals showed a normal preference for the nutritious D-glucose" These adults were maintained at 17°C during their adult stage, what was their rearing temperature? It would be helpful to have that information in the context of Figure 1H.

All flies were reared at 17° during development, which was formerly indicated by “D,”. We now changed the labels from “D” and “A” to “dev” and “adult” for added clarity, and modified the legend.

Figure 1H shows when flies were reared at 17°C and then switched to 31°C at adult stage, the starved animals were unable to make a choice between D- and L- glucose. What happens to the preference when flies are reared at 17°C and examined as adults at 17°C?

These starved flies show normal selection of D-glucose over L-glucose (see bars with blue data points).

Figure 2:Through 5 minutes of GCaMP6s recording, controls and trpγ1/+ have nearly constant oscillation frequency, whereas for trpγ1 mutants the response levels drop over time. Is there an explanation for this? Do we have the data for oscillation number and duration quantified for the GCaMP6s recordings of all 3 genotypes? Also, n is mentioned as 10-20 cells: Does that mean a total of 10-20 Dh44 neurons across flies (knowing there are 6 Dh44 neurons per fly)?

We analyzed less than 6 cells/fly in most cases, since the full complement of Dh44 neurons were typically not in focus in the same focal plane. We tabulated the results from 3—4 animals per genotype to achieve 10—20 cells (added to the legend to *Figure 2C*). We added binned bar graphs showing that the oscillation frequency exhibited by the *trpγ* mutant was similar to the control flies (*Figure 2A, C* and *E*). There also were not a large difference in the rate of decline from the peak fluorescence (*Figure 2B* and *E*; time for 50% decline, *t*_50_=10.1 and 8.2 min for the control and *trpγ^1^*, respectively). This is now described in the Results section (page 10).

Figure 3:Figure 3C legend says n = 6 – 8, but trpγG4 and trpγ1, UAS-trpγ/trpγ1 have 5 data points each for glycogen levels in the 18h starvation period.

We made the correction.

Figure 5:Figure 5C How about using a scatter plot instead of continuous lines for a relatively easier visualization for percentage of flies in each category of crop score?

Due to the complexity of this figure, a scatter plot makes the presentation rather complex to visualize. Therefore, we retained the line graph.

Figure 5- Supplement 1: Line 926 – Male flies were feed 5% of the…. Typo "feed" need correction to "fed".

We made the correction.

Figure 6:Figure 6-Supplement 1: It would be better to add one sentence here stating the method in which the body weight measurements were done.

We weighted groups of 10 males using a digital balance. We added this information to the legend, and to a short section at the end of the Materials and methods.

Reviewer #3:This nice manuscript from Dhakal et al., describes experiments that establish a role for the function of TRP-y in the selection of nutritious foods in hungry animals. In general the paper is well put together and the experiments seem solid. I have a few suggestions for the authors.My main comment relates to the general message in relation to the experiments. The title implies that TRPy might actually be part of the mechanism of food selection but if I follow the work clearly, it seems that actually in most cases the authors conclude that TRPy is required for the proper function of DH44 neurons. It therefore seems that the title needs to be more specific so that it does not mislead readers.

We changed the title to, “*Drosophila* TRPγ is required in neuroendocrine cells for post-ingestive food selection.”

The authors explicitly state on line 375 that 'TRPy is not involved in sensing D-glucose'. The last sentence of the manuscript should also perhaps state that TRPy is functioning in the neurons that are required for taste-independent food-selection, that it functions in the regulation of.

We modified our model as described in our response to the first consensus comment by the reviewers. In addition to functioning partially as a leak channel, we suggest that it also acts to activate the Dh44 neurons, downstream of a glucose-activated receptor. We modified the last sentence of the manuscript as follows,

“Similar to the requirement for TRPγ in Dh44 neuroendocrine cells, which control taste independent feeding in flies, we suggest that TRP channels in neuroendocrine cells in the mammalian brain might also function in the regulation of taste-independent food selection, and in homeostatic control of metabolism, in response to the internal state.”

I also thought the authors could make a better job of fully explaining where things make sense with respect to a role in DH44 neurons and when not. For example, the authors show that the TRPy-GAL4 labels neurons in the crop and they then go on to show phenotypes in the crop. However, they state/find that these phenotypes can be rescued with DH44 neuron expression. This confused me and I am left wondering what the innervation of the GAL4 in the crop means? Is this DH44 neurons? Or are they as the title implies, only in the brain? How specific is the DH44-GAL4 expression to the brain?

Dh44 neurons extend processes from the π that label the gut and crop (Dus et al., 2015). Similarly, we found that the *trpγ* reporter stained neuronal varicosities in the epithelium of the crop and intestine (*Figure 4A* and *B*). Due to co-expression of *trpγ* with Dh44 in the PI, these processes are likely to also extend from the PI. We added this point to page 11.

The authors show that the TRPy-GAL4 expresses in many places in the brain but their study mostly focuses on the DH44 neuron expression. How sure are they that the GAl4 expression fully recapitulates the native expression? I think it would be helpful to mention this and also to comment in more detail on where else the GAL4 expresses. It is notable that defecation is TRPy dependent but not rescued with DH44 or dilp2-GAL4 expression. Is that a brain phenomenon too?

To address our level of confidence in the *trpγ* reporter expression pattern, we added the following (page 9),

“The rescue results, and the ability to phenocopy the *trpγ* deficit by RNAi using the *Dh44-GAL4* support our conclusion that *trpγ* is required and sufficient in Dh44 neurons for proper selection of nutritive D-glucose in hungry flies. Additional support for this conclusion is our finding that the *trpγ* reporter is expressed in Dh44 neurons. The validity of *trpγ* expression in Dh44 neurons is also supported by the design of the *GAL4* driver, which is inserted precisely at the site of the initiation codon (Akitake et al., 2015). However, we do not have independent verification that the reporter fully recapitulates the native expression pattern of *trpγ*.”

Due to broad expression pattern of *trpγ* in the brain, which we display in *Figure 1D*, it would not be easy to accurately describe the rest of the *trpγ* expression pattern without extensive double-labeling experiments. Furthermore, describing the expression pattern outside of the Dh44 neurons is not essential for our conclusion that TRPγ functions in these neuroendocrine cells. Lastly, the CNS can impact on defecation, and we now mention this point (page 16):

“While the relevant neurons remain to be identified, neurons have been described that impact on defecation, some of which are in the central nervous system (Cognigni et al., 2011).”

The physiological effects described in Figure 2 are very interesting and show a clear increased activity in the mutant. The authors provide a plausible explanation for this but they don't show if it has any consequence on release from the DH44 neurons. Would this be possible? Maybe they should at least mention if so?

Please see our response to the second consensus comment from the reviewers. The relevant section is repeated here and appears in the main text (page 20).

“….*trpγ* function is rescued by expressing the wild-type transgene in *Dh44* neurons, and *Dh44* mutant flies exhibited the same deficits in tissue sugar and glycogen levels. These findings raise the possibility that TRPγ promotes the release of DH44, which in turn increases glucose uptake.”

Also, the authors show the peak amplitude is increased in TRPy mutants. Is it correct that the oscillations that follow the initial rise are actually dampened? Is that meaningful, or merely a property of the reporter?

We added binned bar graphs showing that the oscillation frequency exhibited by the *trpγ* mutant was similar to the control flies (*Figure 2A, C* and *E*). Also, there was not a large difference in the rate of decline from the peak fluorescence (*Figure 2B* and *E*; time for 50% decline, *t*_50_=10.1 and 8.2 min for the control and *trpγ^1^*, respectively). This is now described in the Results section (page 10).

Reviewer #4:In this study, Dhakal and colleagues studied the role of Transient receptor potential (TRP) genes in taste-independent nutritional choices in Drosophila. The authors report that TRPgamma expression is required for the ability of (starved) flies to select metabolizable D-Glucose over non-metabolizable yet sweet L-Glucose. Importantly, the authors determined that TRPgamma is expressed in Dh44 neurons, which are known to control taste-independent choices in flies; in fact rescuing TRPgamma expression in Dh44+ neurons was sufficient to induce D-Glucose preferences. The authors do also show that TRPgamma is in addition involved in a number of metabolic and digestive processes.This is an excellent study that greatly advances our understanding of taste-independent preferences. Crucially, the study points to TRP channels as potential regulators of nutrient preferences in mammals, as TRP channels are highly conserved across species. The experiments were all well designed and executed, as well as clearly described.I do have however the following observations. First, it was not mentioned whether the motor control of mouth parts is preserved in TRPgamma mutants. Given that TRPgamma was previosuly shown to be involved in motor control, and in the present study is shown to affect movement of digestive parts, comments from authors are needed to clarify this issue.

We added the following data regarding proboscis extension, which we describe on page 7,

“the time to full extension of the proboscis was similar between the *trpγ^1^* mutant and the control, and we did not detect any obvious motor deficit in proboscis extension by the *trpγ^1^* mutants (*Figure 1—figure supplement 1E*; *video 1* and *2*).”

One other interesting aspect of the study is that TRPgamma appears to be essential for regulating tissue sugar levels (l. 192). This may imply that the reinforcement from that is taste-independent may depend on the organism's ability to metabolize D-Glucose. This idea seems to have some support from mammalian studies (e.g. J Neurosci. 2010 Jun 9;30(23):8012-23) and would therefore be interesting to know whether the authors believe similar processes occur in flies.

This is an interesting possibility. We added the following suggestion to the Discussion (page 19): “In mammals, there is evidence that taste-independent sugar preference is influenced by glucose metabolism (Ren et al., 2010). Thus, it is intriguing to speculate that the deficit in D-glucose selection in *trpγ* mutants is due in part to impairment in metabolism of D-glucose.”

[Editors’ note: further revisions were suggested prior to acceptance, as described below.]

Reviewer #1:The authors addressed most of the concerns and issues raised by me and other reviewers, but it is still unclear to me how a mutation in TRPgamma (a proposed leak channel), which results in hyperpolarization, would "increase the driving force for Ca++ entry following glucose-dependent activation of another Ca++ permeable channel that remains to be identified." As the authors indicated, a knockout of TRPC5 in POMC neurons indeed eliminated leptin-induced depolarization when Gao et al., conducted whole-cell patch-clamp recordings. Contrary to the authors' inference, however, these POMC neurons without TRPC5 do not have increased levels of intracellular Ca++ following leptin treatment.The authors did not perform whole-cell recordings on DH44 neurons in which TRPgamma is knocked out, but I would be very surprised if DH44 neurons without TRPgamma are not depolarized at all in response to glucose treatment. How would glucose-dependent activation of a Ca++ permeable channel be able to endow the driving force for Ca++ entry? Inhibition of the key glucose metabolic pathway in DH44 neurons through pharmacological or mutational manipulations resulted in elimination of glucose-evoked activation of DH44 neuron. Do the authors postulate that such a Ca++ channel in DH44 neurons becomes active when TRPgamma is knocked down? I am confused….There is a precedent where hyperpolarization in pancreatic α cells results in an increase in calcium levels, which leads to glucagon secretion. This hypothesis, however, is without controversy. This manuscript will be benefited enormously if the authors could carry out a physiology experiment to demonstrate the requirement of TRPgamma for the glucose-evoked stimulation of DH44 cells, or at least provide detailed explanations of how hyperpolarization increases intracellular calcium levels in DH44 cells before I become comfortable in recommending eLife to publish this work.

Monitoring changes in ca^2+^ signals is a useful but imperfect proxy for neuronal activation since it is possible to observe a rise in ca^2+^ that is not associated with an increase in neuronal activity. Therefore, we have now added experiments using a genetically-encoded voltage indicator (GEVI), ASAP2s, in conjunction with fast twophoton imaging to record dynamic, rapid changes in voltages. To conduct these experiments, we expressed *UAS-ASAP2s* under the control of the *Dh44-GAL4* in both control and *trp*g*^1^*. The results of these experiments resulted in revision of our previous model. We found that when we bathed the brains in a glucose-free bath, the basal voltage signals were significantly higher in *Dh44* neuroendocrine cells from the *trp*g*^1^* mutant than from control DH44 neurons (*Figure 3A)*. Furthermore, when we switched to a ca^2+^-free buffer, the control *Dh44* cells displayed elevated ASAP2s signals similar to the levels produced in *trp*g*^1^* in a ca^2+^-containing buffer (*Figure 3A)*. These data indicate that ca^2+^ influx is essential to maintain a low, basal level of neuronal activity.

Next, we measured the voltage responses by DH44 neurons in response to glucose. Stimulation of control cells with 20 mM D-glucose increased the peak amplitude 2.3fold relative to the buffer only (*Figure 3B and D*). However, *trp*g*^1^* DH44 neurons exposed to D-glucose, displayed a peak ASAP2s amplitude significantly larger than the control (*Figure 3B, D* and, *F*). In contrast, the non-nutritive L-glucose had no effect on the ASAP2s signal in DH44 neurons from either the control or *trp*g*^1^* (*Figure 3B*). Furthermore, the peak amplitude in *trp*g*^1^* was similar to the responses of the control and *trp*g*^1^* under ca^2+^ free conditions (*Figure 3B*, *3G*). However, the response latencies exhibited by control and *trp*g*^1^* cells exposed to a ca^2+^-containing buffer or ca^2+^-free buffer were not significantly different (*Figure 3C*).

Altogether, these data indicate that ca^2+^ influx through the TRPg channel serves to attenuate rather than increase basal and glucose-stimulated activity of DH44 neurons. Our findings with the GEVI, do not support the previous model suggesting that DH44 neurons in the *trp*g mutant are hyperpolarized. Rather, we propose that similar to TRPC4 in lateral septal neurons (Tian et al., 2014), TRPg functions in DH44 neurons to hold these neurons in an afterdepolarization state, reducing the rate of firing. We suggest that this state underlies the higher peak ASAP2s signal (increased firing rate) and higher baseline activity in *trp*g mutant *Dh44* neurons. We also propose that the TRPg-dependent after depolarization requires extracellular ca^2+^, as is the case for TRPC4. Consistent with this idea, when we measured the ASAP2 signal in control DH44 neurons bathed in a ca^2+^-free buffer, the ASAP2 signals increased, similar to what we observed in the *trp*g mutants bathed in a ca^2+^containing bath. At the physiological level, this would result in the release of excessive DH44, and deplete the neuropeptide stores in these neurons, thereby rendering them unable to respond to a continuous high level of nutrients in the hemolymph. The new data with ASAP2 and the interpretation of the finding are presented in the Results (*Figure 3*) and the Discussion.

Reviewer #2:I appreciate the authors' edits and inclusions. However, they did not address the essential experiments with the exception of the RNAi one. The DH44 secretion experiment and the calcium imaging at a higher concentration need to be done. These experiments do not require additional tools or materials, the authors have done the calcium imaging already and they have the antibodies. In particular, there is not basis of evidence that 2.5mM is the optimum concentration to use or that it is even physiological; thus the phenotype may not be present or be different at higher concentrations.

We performed additional ca^2+^ imaging experiments with 20 mM D-glucose (*Figure 2supplementary figure 1*), and obtained results similar to our previous findings with 2.5 mM D-glucose.

We conducted DH44 secretion experiments by exposing the brains to D-glucose, Lglucose, or D-fructose, and staining with anti-DH44 as previously described (Dus et al., 2015). Consistent with previous findings (Dus et al., 2015), we found that control DH44 neurons exhibited lower anti-DH44 signals when stimulated with nutritious sugars (D-glucose and D-fructose), but not when bathed with the non-nutritious sugar (L-glucose) (*Figure 3—figure supplement 1*). The diminished anti-DH44 staining was more pronounced when stimulated with 2.5 mM and 20 mM D-glucose than with 50 mM D-glucose. Of significance, we found that the anti-DH44 signals were lower in *trp*g*^1^* than in control DH44 neurons, regardless of whether or not the brain was activated by nutritious or non-nutritious foods (*Figure 3—figure supplement 1*). The lower levels even in non-stimulated cells may reflect the higher basal activity of DH44 cells in the *trp*g*^1^* mutant, resulting in greater DH44 release even in the absence of stimulation. These data are presented in *Figure 3—figure supplement 1* and described in the Results.